# Zebrafish as a Useful Model System for Human Liver Disease

**DOI:** 10.3390/cells12182246

**Published:** 2023-09-11

**Authors:** Nobuyuki Shimizu, Hiroshi Shiraishi, Toshikatsu Hanada

**Affiliations:** Department of Cell Biology, Oita University Faculty of Medicine, Yufu 879-5593, Oita, Japan; shiroshi@oita-u.ac.jp

**Keywords:** liver disease, alcoholic liver disease (ALD), drug-induced liver injury (DILI), hepatocellular carcinoma (HCC), NASH/NAFLD, cholestasis, liver regeneration, zebrafish

## Abstract

Liver diseases represent a significant global health challenge, thereby necessitating extensive research to understand their intricate complexities and to develop effective treatments. In this context, zebrafish (*Danio rerio*) have emerged as a valuable model organism for studying various aspects of liver disease. The zebrafish liver has striking similarities to the human liver in terms of structure, function, and regenerative capacity. Researchers have successfully induced liver damage in zebrafish using chemical toxins, genetic manipulation, and other methods, thereby allowing the study of disease mechanisms and the progression of liver disease. Zebrafish embryos or larvae, with their transparency and rapid development, provide a unique opportunity for high-throughput drug screening and the identification of potential therapeutics. This review highlights how research on zebrafish has provided valuable insights into the pathological mechanisms of human liver disease.

## 1. Introduction

The zebrafish (*Danio rerio*), a small fish species that has gained popularity as an ornamental fish, was first discovered in the tributaries of the Ganges River in India in 1822 [1]. The achievements of George Streisinger (University of Oregon, USA) in zebrafish research are significant and have improved our knowledge of genetics, developmental biology, and the use of zebrafish as a model organism. In 1981, Streisinger and his colleague discovered zebrafish mutants that exhibited pigmentation abnormalities and developmental defects after exposure to ultraviolet and gamma radiation. This pioneering research established the foundation for vertebrate developmental genetics and demonstrated the potential of using zebrafish in this field [2]. Since then, zebrafish have also been recognized as useful organisms for research on human diseases. The zebrafish share common organs and 71.4% of the same genes as humans [3], thus making it a widely used animal model in over 3250 institutes across 100 countries for the study of developmental biology, toxicology, cancer, and genetic disorders [4]. In 2013, the zebrafish genome was completely sequenced [3], and the data can be accessed through the Zebrafish Model Organism Database (ZFIN) [5]. Zebrafish are beneficial model organisms for the study of liver development and disease. Its high reproductive rate, transparent embryos and larvae, and rapid development make it ideal for the high-throughput screening of phenotypes related to liver diseases and potential therapeutic drugs. Since the liver is anatomically and functionally complete during the larval stage at 5 days post-fertilization (dpf), creating disease models using larvae allows for easy observation in live animals and saves time [6,7]. The first scientific study of zebrafish was published in 1937 [8]. Fifteen years later, in 1952, a hepatotoxicity study used juvenile zebrafish as a model for human liver disease by adding ethyl carbamate (urethane) compounds [9]. Since then, it has been reported that zebrafish can be exposed to carcinogens, alcohol, high-fat diets, genotoxic mutagens, and toxic viral proteins to mimic various liver diseases observed in humans. These diseases include liver cancer, alcoholic liver disease, cholestasis, fatty liver disease, and viral hepatitis. Furthermore, the advent of CRISPR/Cas9 system [10], which facilitates specific genetic manipulations, has increased the value of using zebrafish as a model to rapidly and robustly characterize the gene functions involved in human liver diseases. In this review, we begin by outlining the structural differences between human and zebrafish livers. Next, we provide an in-depth analysis of the valuable tools and methodologies employed for studying the zebrafish liver and its models of liver diseases such as drug-induced liver injury, non-alcoholic fatty liver disease, cholestasis, and hepatocellular carcinoma (HCC).

## 2. Liver Morphology, Function, and Development in Zebrafish and Humans

### 2.1. Liver Cell Types and Structure

The liver plays a vital role in both humans and zebrafish, and it contains a number of cell types that are responsible for its proper functioning. In humans, the liver consists of hexagonal hepatic lobules, functional units within the liver, that contain hepatocytes, cholangiocytes (biliary epithelial cells), endothelial cells, Kupffer cells (resident macrophages), and hepatic stellate cells. The zebrafish liver contains cell types that are morphologically and functionally analogous to their human counterparts [11,12,13,14,15,16]. Furthermore, a comparative analysis of the transcriptome profiles between zebrafish liver cells, mice, and humans revealed the complete conservation of orthologous cell types in zebrafish that perform identical functions to their human counterparts [17].

The liver structure in teleost fish differs from mammals, which is often described as having “hepatic tubules” rather than mammalian “hepatic lobules” [18,19]. On the other hand, medaka fish (*Oryzias latipes*) exhibit a hexagonal (polyhedral) architecture resembling mammalian livers [20]. Yao and colleagues showed that in the zebrafish liver, the hepatic tubules and sinusoids are arranged in a radial pattern around a central vein [21]. Recently, three-dimensional analyses revealed that zebrafish have a hepatic-lobule-like architecture with polygonal structures. These lobules have central veins at the center, portal veins at the periphery, and connected sinusoids [22]. However, there are some significant differences in the hepatobiliary system and structural organization between zebrafish and humans in the lobules.

In the human liver, bile produced by hepatocytes is transported through bile canaliculi, an intercellular space formed between adjacent hepatocytes, and then through duct-ules, intrahepatic bile ducts, and extrahepatic bile ducts. The ductules and bile ducts are composed of cholangiocytes (biliary epithelial cells). In contrast, in zebrafish, bile is transported from hepatocytes through a mesh-like network of preductules/ductules, which form a tubular lumen with intercellular bile canaliculi on the apical membranes of hepatocytes, and then through the intrahepatic and extrahepatic bile ducts [21,22]. Furthermore, intrahepatic bile ducts in humans are located around the portal vein and hepatic artery, whereas intrahepatic bile ducts in zebrafish are randomly located in the hepatic lobules. This distinction underscores the structural differences in bile transport within the liver lobules (Figure 1).

### 2.2. Liver Zonation

Liver zonation describes how the liver is spatially organized into functional regions with specialized metabolic activities and functions [23]. These include detoxification, metabolism, nutrient storage, and the synthesis of various molecules. These regions are responsible for the detoxification, metabolism, nutrient storage, and synthesis of various molecules. While the zebrafish liver lobule does not have an architecturally distinct zonation, it has been reported that some fishes exhibit zonation after fasting or hepatectomy. Zonation may also occur in zebrafish under specific stress conditions [22]. Researchers have recently performed single-cell RNA sequencing (scRNA-seq) of adult zebrafish livers to study transcriptionally specific populations of hepatic cell types. The gene sets for pathways such as oxidative phosphorylation, lipid metabolism, fatty acid transport, and glucose metabolism suggest the existence of three groups of hepatocytes. Further investigation is necessary to display a zonation similar to that of humans [24].

### 2.3. Cytochromes P450 (CYP)

The liver plays a primary role in catalyzing oxidative transformations that activate or inactivate various internal and external substances. This process involves oxidation, re-duction, and hydrolysis reactions and is primarily mediated by cytochrome P450 (CYP) enzymes in the liver. CYP enzymes are classified into two categories based on their function: one primarily metabolizes endogenous molecules produced within the body, while the other processes xenobiotics, that is, external substances, and possibly some endogenous compounds [25]. This complex metabolic process can be divided into two critical phases: Phase I involves a series of reactions, such as oxidation, reduction, and hydrolysis, which are facilitated by the CYP enzyme system. Subsequently, phase II includes conjugation reactions that complete the metabolic transformation [26].

The selection of an animal model for toxicity testing requires an understanding of the metabolic characteristics of the species that influence drug-induced liver injury (DILI) and compound toxicity. Zebrafish have gained attention for their potential as a model of human hepatic metabolism. Researchers have identified and categorized 94 CYP genes in zebrafish, thereby revealing evolutionary relationships with human CYP enzymes [27]. Zebrafish exhibit metabolic responses similar to those of humans, as have been demonstrated in studies with drugs such as ibuprofen [28]. These experiments confirmed that zebrafish have metabolism and reactive metabolites that are similar to those of humans.

In addition, zebrafish embryos show metabolic phase I and II reactions, including oxidation, N-demethylation, O-demethylation, N-dealkylation, sulfation, and glucuronidation [29]. Studies demonstrated that zebrafish genes for the pregnane X receptor (*pxr*), cytochrome P450 3A (*cyp3a*), and multidrug resistance 1 transporter (*mdr1*) respond to regulation in a manner resembling mammals [30].

Furthermore, the aryl hydrocarbon receptor (*ahr2*), which serves as a receptor for a variety of environmental and endogenous compounds, influences gene expression in zebrafish and regulates phase I and II metabolic enzymes [31]. These studies make zebrafish a promising model for drug metabolism studies and drug-induced liver injury research, thus reflecting human mechanisms.

### 2.4. Liver Development

#### 2.4.1. Endoderm to Hepatic Specification

The endoderm is one of the three primary germ layers that develops during the early embryonic stages. This results in the development of different internal organs and tissues, thus forming the liver, lungs, and pancreas.

In mice, around embryonic day 7.5 (E7.5), the transcription factor *Foxa2* is initially expressed as a pioneer factor during the early endodermal stage [32,33,34]. Subsequently, at E8.5 in the ventral foregut endoderm, *Gata4/6* [35,36,37], *Prox1* [38], and *Hhex* [39,40] contribute to the coordinated regulation of hepatic specification. These transcription factors work together to establish and maintain the identity of the hepatic progenitor cells (hepatoblasts), thereby allowing for precise differentiation into functional liver cells.

In zebrafish, knockdown and overexpression experiments have indicated that these transcription factors (*foxa2*, *gata4/6*, *prox1*, and *hhex*) function similarly to their role in endoderm formation and hepatic specification in mammals [41,42,43].

#### 2.4.2. Hepatobiliary Differentiation

The hepatoblasts, around 24 hpf, form a distinct bud-like structure from the left side of the zebrafish embryo. In mice, the hepatic bud receives critical inductive signals from the cardiac mesoderm, endothelium, and septum transversum mesenchyme (STM) [44]. The presence of the STM in zebrafish raises doubts, with indications that the lateral plate mesoderm (LPM) may fulfill a comparable role in this species [45]. During liver development, bipotential hepatoblasts differentiate into hepatocytes and cholangiocytes (biliary epithelial cells: BECs), which form the intrahepatic and extrahepatic bile ducts. The process of hepatocyte differentiation in zebrafish is governed by a complex network of signaling pathways, such as Wnt, Fgf, and Bmp, which respond to external stimuli to ensure controlled growth and differentiation [46,47,48,49].

#### 2.4.3. Hepatobiliary Outgrowth

Liver growth in zebrafish occurs through dynamic processes of hepatobiliary cell survival and proliferation. Genes, such as *anxa4*, *snx7*, and *id2a*, play a central role in this regulation. The survival of hepatoblasts is ensured by *anxa4* and *snx7* [50]. On the other hand, *id2a* plays a crucial role in the proliferation of hepatoblasts [51]. The *snapc4*, another gene, is critical for the survival and proliferation of cholangiocytes, thus contributing to the diverse aspects of liver development [52]. Genes related to mitochondria, including *tom22* [53], *subv3l1* [54], and *trx2* [55], which are essential for hepatocyte survival, highlight the significance of energy homeostasis in maintaining liver growth. Furthermore, chromatin remodeling factors such as *hdac1*, *hdac3*, *ssrp1a*, and *uhrf1* play a crucial role in governing the cell growth and cell cycle [56,57,58,59,60]. Deficiency or compromised activity of these chromatin remodeling factors usually results in a reduced liver size by 5 dpf. This is mainly due to a decrease in cell proliferation and interruptions in DNA replication. Insights from zebrafish studies and the comparative understanding across mammalian models increase our comprehension of liver growth regulation. These mechanisms can positively affect liver health and regenerative strategies across different species.

#### 2.4.4. Key Signaling Pathway in Liver Development

Wnt signaling stimulates hepatic growth and maturation in zebrafish and mice. The manipulation of Wnt expression at various stages affects liver specification. The loss of *apc*, a key component of the Wnt pathway, promotes liver expansion at the expense of the pancreas [46]. Furthermore, the importance of Wnt is highlighted by the impaired hepatoblast formation of the *prometheus (prt)* mutant, whose mutation occurs in the *wnt2bb* gene [45]. Wnt signaling, which is governed by *hnf1b* and *epcam*, initiates during hepatic specification, thereby promoting hepatoblast proliferation after specification [61,62].

Fgf signaling plays a critical role in hepatic specification in both mice and zebrafish. In zebrafish, inhibition of the Fgf receptor between 18 and 26 hpf leads to reduced expression of *prox1*, *hhex*, and *gata4/6* at a later stage [43,63]. Among the members of the Fgf family, *fgf10* from the mesenchyme prevents hepatocytes from differentiating into other organs and tissues [64]. Nevertheless, it remains uncertain whether the mesenchyme is an exclusive source of the Fgfs that are necessary for zebrafish hepatic specification.

Although Bmp signaling plays an essential role in mammalian hepatic differentiation, zebrafish, which lack an STM, demonstrate the importance of Bmps in hepatic specification. Mutants lacking the Bmp/activin receptor and embryos with dominantly negative Bmp receptor overexpression showed reduced hepatic gene expression [43]. The Bmp requirement decreases with the embryonic stage, thus corresponding to the period when Fgfs are necessary. Genetic screening revealed an unexpected role of myosin phosphatase targeting subunit 1 (*mypt1*) in LPM positioning and hepatic induction by *bmp2a* [65].

The Notch signaling pathway has a recurring effect on biliary system development. Interfering with Notch signaling in the early stages of development disrupts the formation of cholangiocytes, which is similar to the phenotypes associated with *jagged1* in Alagille syndrome [6]. Notch signaling coordinates the remodeling of cholangiocytes into a functional network, which requires the action of *sox9b*, the target of Notch signaling [66]. A deficiency in *sox9b* leads to irreversible malformations in the zebrafish biliary system, thereby resulting in cholestasis [67]. The transcription factors *onecut1/hnf6*, *onecut3*, and *vhnf1* play pivotal roles in the development of the zebrafish hepatobiliary network, which mirrors that of mammals [68,69]. Cell polarity and tight junctions play critical roles in the development of the mammalian hepatobiliary system [70], as has been confirmed by zebrafish claudin15-like b mutations that cause biliary anomalies [71]. Furthermore, the regulation of biliary network maintenance by *snapc4* highlights its specificity for biliary cells [52]. Integrating insights from zebrafish with rodent models will enrich our understanding of liver patterning, fate determination, and growth control. This holistic approach parallels similar paradigms in liver regeneration research.

## 3. Useful Tools for Analysis of Zebrafish Liver

### 3.1. Forward Genetics

Forward genetics is an approach in genetic research that involves identifying the genes responsible for a particular phenotype or trait without prior knowledge of the function of the gene. The understanding of organ formation and genetic diseases through large-scale forward screens has rapidly made zebrafish the primary model organism in biomedical research.

Forward genetics is based on a phenotype-driven approach that induces random mutations in zebrafish using mutagens, such as N-ethyl-N-nitrosourea (ENU) or retroviral insertions. This creates a diverse population of mutants with varying phenotypes. Subsequently, researchers identify the phenotypes of interest and perform positional cloning to identify the responsible genes. This process utilizes genetic markers and linkage analysis and is time-consuming. However, the development of next-generation sequencing has accelerated the identification of responsible genes. ENU introduces single-nucleotide changes that frequently cause loss-of-function alleles and subsequent phenotypic changes. Retroviral mutagenesis is the process of inserting retroviruses into the genome, thereby disrupting genes and generating diverse phenotypes [72]. Despite these differences, these techniques effectively facilitated the discovery of crucial development-related genes.

In the mid-1990s, two main groups, Christiane Nüsslein-Volhard (Max Planck Institute, Germany) and Wolfgang Driever (Massachusetts General Hospital, USA) played a crucial role in the large-scale screening of zebrafish, which resulted in a variety of phenotypically diverse mutants. The outcomes were published in a special issue of the Development Journal [Development 123,1-460 1996] [7,73,74].

Despite being time-consuming, many laboratories have successfully applied this ENU-induced mutagenesis to inactivate several liver development-related genes in zebrafish. The *wnt2bb* mutant exhibits compromised liver development, reduced hepatocyte proliferation, and impaired bile duct formation [45]. The *uhrf1* mutant displays disrupted hepatic outgrowth and liver regeneration capacity after injury, along with altered DNA methylation patterns [60]. To gain a better understanding of other liver mutants, we recommend reading a research article by Jaime Chu et al. [75], which offers an extensive list of information.

Forward genetics is an unbiased method that avoids preconceived notions of target genes and reveals the relationship between genes and human diseases. This process reveals subtle roles and even redundant functions gene functions that are often missed by reverse genetics, as discussed below, and it provides insight into human diseases and new therapeutic targets to understand liver diseases. It is essential to identify the responsible genes and signaling pathways; however, many aspects remain unknown. Only a few genes have been reported to regulate bile duct formation in the liver [76,77]. Forward genetic screens could uncover the genes involved in this process.

Recently, using ENU mutagens, 24 zebrafish mutants that exhibited cholestatic liver injury due to abnormal bile duct formation were identified. Using next-generation sequencing, one of these mutants, *nckap1l^lri35^*, was found to have a mutation that results in an early termination codon in a minor splice isoform of the *nckap1l* gene. It is not currently known whether a mutation similar to *nckap1l^lri35^* exists in humans. However, based on the sequence data obtained from patients with biliary stasis, the human *NCKAP1L* gene should be screened for mutations [78]. In addition, most of the other zebrafish mutants discovered in this study survived into adulthood, thereby providing a valuable new animal model for chronic human cholestasis and providing new insights into this disease.

In conclusion, forward genetics continues to be a powerful and relevant tool for zebrafish research because of its capacity to uncover novel phenotypes, identify essential genes, leverage technological advancements in sequencing, and contribute to the study of human liver diseases.

### 3.2. Reverse Genetics

Historically, the disruption of target genes (null or strongly hypomorphic alleles) in zebrafish has been more difficult than conducting that process in mice. Zebrafish lack established embryonic stem (ES) cell lines that allow for homologous recombination with targeting vectors, which is a useful technique for mice. This restricts the application of genetic recombination approaches.

#### 3.2.1. TILLING System

In 2003, Cuppen and his coworker applied the Targeting Induced Local Lesions IN Genomes (TILLING) method to disrupt target genes in zebrafish [79,80]. The TILLING method is a powerful method used in zebrafish research to identify and characterize specific point mutations in targeted genes. Zebrafish males are exposed to the ENU mutagen, and their sperm are collected and frozen to create a diverse mutant library. Researchers then sequence the genomic DNA from this large library of target genes to identify mutations. This approach allows for the comprehensive establishment of zebrafish strains with various mutations in specific genes [81]. To identify loss-of-function mutations, it is desirable to screen the 5′ exons intensively. This is because mutations causing a premature stop codon in the first exon may result in a nonfunctional protein. The TILLING method can also be used to identify novel missense mutations that fail to fully complement the null mutations. The discovery of such hypomorphic alleles is a great advantage of ENU mutagenesis in zebrafish compared with gene-targeting strategies in mice, which most often generate null alleles.

#### 3.2.2. ZFN System

The first generation of engineered DNA nucleases, zinc finger nucleases (ZFNs), offers targeted genome editing through zinc finger motifs that bind to specific DNA sequences [82]. Comprising DNA-binding zinc finger domains and FokI nuclease domains, ZFNs create double-strand breaks in the desired DNA sequences. These breaks trigger cellular repair mechanisms, thereby enabling precise gene editing through either error-prone nonhomologous end-joining (NHEJ) or template-guided homology-directed repair (HR). Their ability to induce site-specific mutations was initially promising in zebrafish research [83]. However, the ZFN system has several drawbacks, including a labor-intensive design, limited target flexibility, high cost, and off-target effects. Therefore, the use of ZFNs has not become widespread in other model organisms, including zebrafish.

#### 3.2.3. TALEN System

The transcription activator-like effector nuclease (TALEN) provides an adaptable alternative that utilizes customizable DNA-binding domains for precise genome editing [84]. The TALEN consists of customizable TALE repeats for DNA-binding and a FokI cleavage domain for targeted double-strand breaks (DSBs). These breaks activate repair pathways, including the NHEJ, thus leading to gene knockout and the HR allowing controlled genetic modifications. This application in zebrafish research has been explored, thus emphasizing their contributions to generating diverse genetic modifications [85]. The use of dsDNA donors enhances HR efficiency, thereby allowing researchers to insert specific sequences into zebrafish genomes [86]. However, the design and implementation of CRISPR-Cas9 experiments are relatively straightforward, which require less time and effort than traditional methods that require elaborate genetic manipulations.

#### 3.2.4. CRISPR/Cas9 System

The clustered regularly interspaced short palindromic repeats (CRISPR) RNA-guided Cas9 nucleases (Cas9s) provide a revolutionary genome editing tool that offers distinct advantages over its predecessors, the ZFN and TALEN systems. Unlike ZFNs and TALENs, the CRISPR/Cas9 system does not require the intricate design of custom proteins for each target gene, thereby making it more versatile and cost-effective. The history of CRISPR/Cas9 dates back to the discovery of its role in bacterial immunity. In 2012, Doudna and Charpentier harnessed this natural mechanism to develop a powerful genome-editing tool [87]. The CRISPR/Cas9 system employs a single-guide RNA (sgRNA) to guide the Cas9 enzyme to a specific DNA sequence. The sgRNA is a synthetic fusion of CRISPR RNA (crRNA) and transactivating CRISPR RNA (tracrRNA), which simplifies the editing process [88].

The CRISPR/Cas9 system has been successfully applied for genome editing in zebrafish [89]. Targeted genetic modifications can be achieved by injecting either Cas9 mRNA or the Cas9 protein, along with sgRNA, into zebrafish embryos. Cas9 mRNA directs embryonic cells to produce the Cas9 enzyme, whereas delivering the Cas9 protein directly ensures immediate activity. Zebrafish embryogenesis occurs rapidly, thus making timely and efficient gene manipulation crucial. The use of the Cas9 protein for gene editing in zebrafish is important for optimizing editing efficiency.

CRISPR/Cas9’s incredible precision sometimes comes with off-target effects, such as unintended alterations to non-targeted DNA sequences [90]. Interestingly, F0-based genetic assays generally exhibit lower off-target rates (1.1–2.5%) in zebrafish embryos, thus making them a potent tool for precise in vivo genome engineering [91]. To address this concern, Cas9 variants such as D10A have been developed to minimize off-target effects [92], although they may have a lower editing efficiency. In addition, by mating edited zebrafish with wild-type zebrafish, the offspring inherit a combination of edited and wild-type alleles. This genetic dilution reduces the frequency of off-target mutations in the subsequent generations.

### 3.3. Transgenesis

In 1988, the first transgenic zebrafish was generated by the microinjection of plasmid DNA containing exogenous genes into fertilized eggs [93]. Subsequently, in 1995, a zebrafish line that stably expressed GFP was reported [94]. The creation of transgenic zebrafish is an essential technique for analyzing gene function and visualizing cells. However, these methods often result in low success rates, and they are time-consuming.

#### Tol2 Transposon System

The Tol2 transposon system developed by Kawakami (National Institute of Genetics, Japan) and his colleagues has drastically changed this situation. They demonstrated that the Tol2 element in the genome of medaka fish (*Oryzias latipes*) is functionally active in zebrafish and serves as a tool for efficient gene transfer [95,96]. The Tol2 transposon system is based on a transposase enzyme that recognizes specific sequences on both minimal cis sequences of the Tol2 element (~200 bp) and zebrafish genome DNA. By coinjecting the synthetic Tol2 transposase mRNA and Tol2 cis elements into zebrafish embryos, researchers can promote the integration of transposon DNA at specific sites in the genome. [97]. The Tol2 system enables the precise and efficient integration of transgenes, thereby leading to consistent expression patterns and the rapid generation of stable transgenic lines. This advancement has greatly accelerated research on zebrafish, thus facilitating investigations into developmental biology, disease modeling, and gene function. The Tol2 transposon system is known for its high germline transmission efficiency (50–70%), which is a key factor in the successful establishment of stable transgenic lines in zebrafish [98,99]. The Tol2 system has many other advantages, including the reduction of the gene-silencing effect [100], sustained transgene expression over time, and accommodating substantial insertion sizes for versatile genetic modifications [101].

### 3.4. Visualization of Distinct Cell Types in the Zebrafish Liver

Fluorescence-based visualization of the cell types under gene promoters in zebrafish provides critical insights into developmental and molecular processes. By selectively tagging specific cells with fluorescent markers, researchers can pinpoint their location, track their development over time, and uncover how genes regulate their behavior. This method helps to understand complex cell interactions, unravel gene networks, and study disease mechanisms. The transparent nature of zebrafish embryos allows for the non-invasive observation of internal structures, thereby increasing the accuracy of spatial and temporal analyses. Ultimately, this approach provides a platform for studying gene function, cell lineage, and disease modeling, thus advancing our understanding of fundamental biological phenomena and potential therapeutic interventions. To date, the visualization of individual cell types comprising the zebrafish liver, including hepatocytes, endothelial cells, biliary epithelial cells, macrophages, and hepatic stellate cells, has been achieved using Tol2 transposon-mediated transgenic techniques [95,97,102] (see Table 1).

#### 3.4.1. Hepatocytes Marker (*fabp10a*)

Hepatocytes, the major functional cells of the liver, play a central role in liver pathophysiology. The *fabp10a* (also known as *L-fabp*) promoter [103] was used to selectively target zebrafish hepatocytes. The *fabp10a* gene encodes the fatty acid binding protein 10a, which is highly expressed in zebrafish hepatocytes. Integration of the *fabp10a* promoter with various genetic reporters or fluorescent proteins [104] allows for the visualization and observation of processes that are specific to hepatocytes, such as liver growth, regeneration, metabolic disorders, and responses to toxic insults. In addition, the strategic integration of cancer driver genes downstream of the *fabp10a* promoter presents a promising avenue for the development of hepatocellular carcinoma (HCC) models (see Section 6 and Figure 2).

#### 3.4.2. Biliary Epithelial Cells Marker (*keratin18* and *notch1*)

Biliary epithelial cells (cholangiocytes) line the biliary tree and contribute to liver diseases such as cholangiopathies. The *keratin 18* promoter [13] has proven effective in labeling and studying these cells in zebrafish. By coupling the *keratin 18* promoter with fluorescent proteins, researchers can examine the behavior of biliary epithelial cells during biliary development, bile duct formation, and pathological processes, such as cholestasis. Additionally, the utilization of a notch-responsive reporter *tp1bglob* enables the monitoring of biliary epithelial cells and Notch signaling activation [12], which is a critical factor in determining the fate of biliary cells and promoting regeneration.

#### 3.4.3. Endothelial Cells Marker (*flk1* and *fli1*)

Endothelial cells form an intricate network of blood vessels in the liver that facilitate nutrient exchange, detoxification, and oxygenation. To study endothelial cells in zebrafish, researchers have used *flk1* [105] and *fli1* [106] promoters. These promoters drive the expression of endothelial-specific genes, thereby allowing the visualization of vascular development, angiogenesis, and pathological angiogenesis in liver diseases, such as cirrhosis and hepatocellular carcinoma [107].

#### 3.4.4. Hepatic Stellate Cells Marker (*hand2*)

Hepatic stellate cells play a pivotal role in liver fibrosis, which is a common response to chronic liver injury. The *hand2* [106] promoter has been employed to target and investigate hepatic stellate cells in zebrafish. By incorporating fluorescent markers driven by the *hand2* promoter, researchers can explore hepatic stellate cells’ activation, migration, and fibrogenesis, thereby shedding light on the cellular and molecular mechanisms underlying liver fibrosis.

#### 3.4.5. Kupffer Cells Marker (*mpeg1*)

Kupffer cells are a specialized type of immune cells found in zebrafish. The *mpeg1* promoter has been used to visualize and investigate Kupffer cells in zebrafish [108]. They are equivalent to macrophages in mammals and play a crucial role in the immune system of zebrafish. Kupffer cells are predominantly located in the liver, where they are strategically positioned to encounter and eliminate pathogens and foreign particles that enter the bloodstream.

Furthermore, Table 1 includes methods such as antibody labeling for cell types and subcellular structures, various histological staining techniques for liver pathology evaluation, and an array of biochemical assays to examine liver functions.

**Table 1 cells-12-02246-t001:** Useful tool and analysis for zebrafish liver analysis.

Transgenic Zebrafish Lines Labeling Liver Cell Types		
Transgenic Line	Labeling Cell Types	Ref
*Tg(fabp10a:GFP)* *Tg(fabp10a:dsRed) ^gz15^*	Hepatocytes	[104,109]
*Tg(tp1:EGFP)^um14^, also known as* *Tg(EPV.TP1-Mmu.Hbb:EGFP)*	Cholangiocytes (Biliary epithelial cells: BEC), intrahepatic biliary cells	[12,110]
*Tg(krt18:EGFP)*	Cholangiocytes intrahepatic biliary cells, extrahepatic biliary cells, gallbladder	[13]
*Tg(flk1:gfp) ^843^* *Tg(flk1:ras-Cherry)^s896^*	Endothelial cells	[111,112,113]
*Tg(fli1:EGFP) ^y1^*	Endothelial cells	[106]
*TgBAC(hand2:EGFP)*	Hepatic stellate cells	[15,114]
*Tg(wt1b:EGFP)*	Hepatic stellate cells	[115]
**Histological Staining and Analysis**		
**Histology**	**Usage and Note**	**Ref**
Hematoxylin and Eosin (H and E)	Overall tissue morphology, cell types, and basic structures Optimal fixation: Dietrich’s fixative > 10% formalin > 4% PFA	[116]
Oil red O	Detect neutral lipids and triglycerides/Steatosis Advantage: low costs and simple procedure	[117]
Nile red fluorescence	Detect neutral lipids/lipid droplets/Steatosis Advantage: low costs and simple procedure	[118]
Masson’s Trichrome	Assess fibrosis and collagen Differentiates collagen (blue), nuclei (dark brown/black), and cytoplasm (red)	[119]
Periodic Acid Schiff (PAS)	Detect glycogen content	[120,121]
**Antibodies Labeling Liver Cell Types and Intracellular Structure**		
**Antibody**	**Labeling Cell Types and Structure**	**Ref**
Prox1	Hepatoblasts, hepatocytes, intrahepatic biliary cells	[14,64]
Hnf4a	Hepatocytes	[64]
Alcam	Cholangiocytes (biliary epithelial cells) network	[14]
Annexin A4 (Anxa4)/2F11	Cholangiocytes network	[50,64,122]
Cytokeratin18 (Krt18)	Cholangiocytes network	[69,110]
Abcb11/BSEP	Bile canaliculus	[14,123]
Mdr1	Bile canaliculus	[110,117,123]
Desmin	Hepatic stellate cells	[15]
GFAP/Glial fibrillary acidic protein	Hepatic stellate cells	[15]
**Fluorescent Probes for Lipid Metabolism in Liver**		
**Molecular Probes**	**Usage and Note**	**Ref**
PED6	Quenched phospholipid (BODIPY-labeled phospholipase A2 substrate) Visualization of lipid metabolism (lipid uptake and transport) and digestive organ morphology. Screening for lipid metabolism mutants. Monitor for hepatobiliary toxicities.	[124,125,126,127]
BODIPY 493/503	Fluorescent fatty acid analogue (BODIPY-conjugated fatty acid) Visualization of lipid droplets	[128]
BODIPY FL C5	Fluorescent medium-chain fatty acid analogue (BODIPY-conjugated fatty acid) Visualization of lipid transport and metabolism (bile secretion of hepatocytes and bile conduction) Digestive organ morphology (bile ducts and gallbladder)	[128,129]

## 4. Drug-Induced Liver Injury (DILI) Model

Drug-induced liver injury (DILI) is a major concern in drug development. Typically, primary human hepatocytes or the human hepatocellular carcinoma cell line HepG2 in 2D culture are employed as surrogate liver models for studying drug hepatotoxicity [130]. However, these models have limitations, such as diminished liver function and difficulties in achieving a long-term culture. In recent years, advancements in tissue engineering techniques have enabled the development of liver organoids [131], which allow for the construction of three-dimensional (3D) liver tissue comprising various cell types, such as liver parenchyma and bile excretory channels. These techniques offer a comprehensive means of analyzing the metabolic functions of human organs. Nonetheless, 3D culture methods are intricate and require expertise and resources to achieve a consistent quality.

The zebrafish liver performs similar functions as the human liver, including bile secretion, glycogen and lipid storage, insulin response, ammonia metabolism, and the secretion of serum proteins such as complement, coagulation factors, and albumin-like proteins [132,133]. Prominent pharmaceutical companies have conducted studies to demonstrate the added value of zebrafish in predicting drug-induced liver injury in humans. Drug-induced liver injury between zebrafish and humans has been demonstrated using hepatotoxic over-the-counter drugs, including acetaminophen [134,135], tetracycline [136,137,138], erythromycin [137,139], aspirin [137], amiodarone [140,141,142], and cyclosporine A [132,143] (see Table 2).

Furthermore, zebrafish have been used to identify compounds that mitigate hepatotoxicity caused by the antidepressant trazodone and as new therapeutic agents for liver cancer [144]. Importantly, all the evaluated liver functions were validated in zebrafish within 7 dpf. Additionally, drug hepatotoxicity can be tested using multiwell plates in vitro, thereby eliminating the need for oral administration or intravenous injection in mammalian animals. By simply exposing swimming zebrafish to the drug, data on both the liver and the individual can be obtained simultaneously [145]. Consequently, zebrafish serve as a valuable surrogate model for the human liver, thus enabling the validation and demonstration of hepatotoxicity induced by known and novel drugs, as well as environmental contaminants.

## 5. Alcoholic Liver Disease (ALD) Model

Alcoholic liver disease (ALD) is a significant health issue that causes morbidity and mortality worldwide [146]. Zebrafish larvae exposed to ethanol after liver development show acute effects, such as behavioral changes, liver enlargement, and steatosis [147]. These effects are reversible within 24 h, thus suggesting the possibility of therapeutic insights. Adult zebrafish show similar responses to liver injury, thereby allowing the effects of ethanol to be tested at different concentrations and durations [148,149,150]. However, the use of adult zebrafish for long-term ethanol exposure studies is challenging owing to the limited availability of experimental animals. Moreover, maintaining prolonged exposure to high concentrations of ethanol is complicated because intoxicated larvae struggle to maintain a sufficient food intake. Therefore, optimal research strategies utilize zebrafish larvae to study acute ethanol effects, thus contributing to a better understanding of the physiological impact, metabolism, and regeneration of the liver in a zebrafish model.

Ethanol-induced triglyceride accumulation is associated with increased lipogenesis, restricted lipid utilization, and impaired lipoprotein export [151]. Zebrafish reflect this by linking sterol regulatory element-binding protein (*srebp*). Furthermore, ALD models in zebrafish show reactive oxygen species (ROS) in steatosis [147]. Alcoholic patients exhibit serum protein deficiency due to impaired hepatic secretion [152], which is mirrored in zebrafish by the activation of the unfolded protein response (UPR) to ER stress after ethanol exposure. An important finding is that genetic and RNA-seq analyses indicate that the transcription factor Atf6, which mediates the UPR, plays a central role in the development of alcoholic fatty liver disease [153].

Ethanol exposure activates stellate cells and fibrotic processes. Transgenic zebrafish expressing GFP in the hepatic stellate cells *Tg(hand2:GFP)* facilitate the study of extracellular matrix (ECM) deposition [15]. Zebrafish that are genetically modified to specifically ablate hepatocytes show increased fibrosis after ethanol treatment [154,155]. This comprises quick production of the extracellular matrix (ECM), steatosis, and hepatocyte modifications. Despite not leading to advanced fibrosis, this method offers a valuable model for the development of antifibrotic treatments.

## 6. Hepatocellular Carcinoma (HCC) Model

According to the global cancer statistics published in 2023 [156], hepatocellular carcinoma (HCC) ranks as the third leading cause of cancer-related deaths and the sixth most common primary cancer worldwide. In the United States, approximately 0.6 million individuals succumb to liver cancer annually [157]. HCC, the predominant form of liver cancer, exhibits poor responsiveness to chemotherapy, and it is associated with adverse effects [158]. Recent advancements in genomic analysis technology, coupled with the remarkable development of AI-based drug discovery platforms utilizing AlphaFold protein structure models, have facilitated the identification of novel genetic mutations and therapeutics for HCC [159]. Animal models play a crucial role in investigating the causality of these genetic mutations in liver cancer and in assessing the effectiveness of potential therapeutic candidates. In addition, because early-stage liver cancer in humans is asymptomatic and difficult to detect, animal models are an important tool for establishing more accurate early diagnosis methods. Through the examination of these models, it becomes feasible to explore the interplay between the liver and other organs in the context of liver cancer development. In recent years, researchers have successfully developed zebrafish models of hepatocellular carcinoma that recapitulate the key features of human liver cancer. Transcriptome analysis indicated that the characteristics and progression of molecular interactions in liver cancer are highly conserved between humans and zebrafish [160,161].

Mutations in the adenomatous polyposis coli (APC) gene, a key regulator of the Wnt pathway, are frequently associated with HCC. The TILLING approach (see Section 3.2.1) successfully generated zebrafish carrying *apc* mutations similar to those observed in HCC patients [162]. Molecular analysis revealed aberrant activation of the Wnt pathway in mutant zebrafish, thus mimicking the signaling seen in HCC.

Based on current knowledge, a range of molecular pathways contributes to HCC. Therefore, distinct therapeutic strategies are required for different HCC subclasses due to genetic variations. To develop these strategies, diverse HCC models are essential. Transgenic zebrafish lines with various oncogenes that reflect human HCC have already been established. Additionally, studies have reported the successful xenotransplantation of human hepatocytes into zebrafish, thereby enabling comprehensive in vivo analysis of cancer behavior, gene profiles, and metabolism through imaging techniques and omics analysis [163]. In the following section, we summarize the findings from transgenic and xenograft HCC zebrafish models.

### 6.1. HCC Transgenic Models

Transgenic zebrafish models have been instrumental in studying the contribution of genetic mutations to HCC. By introducing oncogenic driver genes with human HCC, researchers have successfully generated transgenic zebrafish that develop liver tumors mimicking HCC. By coupling the *fabp10a* promoter with oncogenes that are relevant to HCC, researchers can manipulate gene expression specifically in hepatocytes, thus recapitulating key molecular events associated with HCC initiation and progression (Figure 2).

#### 6.1.1. Constitutive Gene Expression System

Transgenic zebrafish are a vital tool in liver cancer research and are engineered to advance our understanding of disease mechanisms. The *fabp10a* promoter drives the constitutive expression of *kras^V12^*, a mutant form of the KRAS gene implicated in cancer [164]. This manipulation activates the MAPK and Wnt/β–catenin pathways, thus mimicking the key pathways in human liver cancer. In addition to KRAS, numerous transgenes that induce HCC have been reported, including Xenopus *Ctnnb1/β-catenin* [165,166], *UHRF1-GFP* [166,167], *HBx-mCherry* [168], and *edn1* [169] (Figure 2A).

However, these transgenic models have several disadvantages. One drawback is that *Tg(fabp10a:kras^V12^)* transgenic zebrafish experience premature deaths prior to adulthood [164]. The uncontrolled activation of oncogenes leads to excessive cellular proliferation and impaired cellular homeostasis, thus resulting in the premature deaths of transgenic zebrafish. Additionally, the constitutive nature of oncogene expression may not accurately mimic the sporadic and multistep nature of human liver cancer development, which typically involves a series of genetic and epigenetic alterations. Several strategies can be considered to overcome the disadvantage of the low survival rates in *kras^V12^* constitutively expressing transgenic zebrafish.

#### 6.1.2. Mifepristone-LexPR and Tetracycline-Tet-On Systems

Gong et al. established two inducible systems, mifepristone-LexPR and tetracycline-Tet-on, to control oncogene expression in the livers of transgenic zebrafish. Mifepristone treatment leading to the overexpression of *krasv^12^* led to dose-dependent tumor induction, thus activating the Raf-MEK-ERK pathway [170]. Liver cancer can also be conditionally induced by other cancer driver genes such as *tgfβ1a* [171], as well as zebrafish *myca* and *mycb* [172]. A mifepristone-inducible line overexpressing *tgfβ1a* revealed its role in NASH-to-HCC transformation, thus inducing HCC and cholangiocarcinoma (CCA) in a dose-dependent manner [171]. This comprehensive approach using zebrafish models helped elucidate the mechanisms of hepatocarcinogenesis involving diverse oncogenes and pathways (Figure 2B).

The Tet-on system [173] induced rapid liver enlargement and tumorigenesis, which often progressed to hepatocellular carcinoma (HCC) with various oncogenes such as *CTNNB1mt* and *tcf7l2* [174], *kras^V12^* [175], *HCP* [176], *xmrk* [177], and mouse *Myc* [178].

These systems enable the temporal control of tumor initiation and progression, thereby providing insights that are not feasible for human patients (Figure 2C).

#### 6.1.3. Mifepristone–Cre/loxP System

Previous inducible transgenic models required continuous inducer exposure, thereby leading to uniform liver cell transformation, unlike human cancer. In addition, continuous chemical exposure interferes with physiological conditions or subsequent drug use experiments. Researchers introduced a novel approach combining mifepristone-inducible *kras^V12^* and Cre/loxP systems [164]. Controlled by the *fabp10a* promoter, *kras^V12^* featured a loxP-flanked mCherry-stop cassette. The induced Cre recombinase excised the cassette with mifepristone, thus inducing permanent *kras^V12^* expression. Although some recombination was incomplete, the resulting liver tumor nodules lacked mCherry fluorescence, thereby resembling clonal human liver tumor proliferation. This innovative strategy employs short-term, targeted chemical exposure to endure genomic changes, thus presenting a more faithful hepatocarcinogenesis model (Figure 2D).

#### 6.1.4. Tamoxifen-Inducible System

Prior transgenic models overexpressing single oncogenes induced primary liver cancer without clear metastasis, possibly because of high severity and short survival. To address this, researchers have introduced a *Twist1a-ERT2* transgenic line in which epithelial-mesenchymal transition (EMT), a metastasis-initiating process, is regulated by Tamoxifen [179]. Combining the *Twist1a-ER^T2^* and *xmrk* oncogene double transgenic model showed extensive hepatic cell dissemination from the tumor liver, distant migration, and secondary tumor establishment. These findings underscored the interplay between oncogenes and EMT genes in initiating metastasis in zebrafish models, thereby offering novel insights into the metastatic cascade (Figure 2E).

#### 6.1.5. CRISPR/Cas9 System

Metastasis research has focused on Dishevelled-associated antagonists of beta-catenin 2 (Dact2), which acts as a tumor suppressor. In zebrafish, *dact2* knockout using the CRISPR/Cas9 system results in increased expression of genes such as *mmp2* and *mmp9*, which are associated with invasion and migration. In addition, *dact2* mutants showed that the epithelial–mesenchymal transition (EMT) transcription factors snail, VEGF, and ZEB were profoundly altered. These alterations affect the liver and other organs by inducing bile duct proliferation and pancreatic invasion. The study highlights the critical role of *dact2* in EMT gene regulation and tumor progression in zebrafish, and it provides insight into these mechanisms through the knockout model [180].

### 6.2. Hepatitis B Virus (HBV) and Hepatitis C Virus (HCV) Model

Hepatitis B virus (HBV) and hepatitis C virus (HCV) have the potential to directly disrupt cancer-related genes and contribute to the development of hepatocellular carcinoma (HCC) [181]. The persistence of HBV or HCV infection results in inflammation, oxidative stress, fibrosis, and cirrhosis [182]. Intriguingly, viral core proteins, such as HBx and HCV, induce dysregulation of cellular processes, thereby promoting the development of HCC [183,184]. Through the utilization of transgenic zebrafish models, it has been observed that HBx expression elicits steatosis and liver hyperplasia, whereas hepatitis C virus core (HCP) expression increases the occurrence of HCC upon exposure to carcinogens. Remarkably, the simultaneous coexpression of HBx and HCP led to the manifestation of severe liver fibrosis and intrahepatic cholangiocarcinoma. Therefore, continued advancements in the development of innovative zebrafish models are of paramount significance in unraveling the intricacies of HCC mechanisms and devising targeted therapeutic strategies.

### 6.3. High-Fat Diet (HFD) Induced HCC Progression Model, and Therapeutics

A HFD worsens liver inflammation and cancer progression in a zebrafish model of nonalcoholic fatty liver disease (NAFLD)/nonalcoholic steatohepatitis (NASH)-associated HCC [107]. Metformin, a controversial treatment for NAFLD and HCC, affects the immune response by altering macrophage polarization and T cell infiltration. It reduces liver size and micronucleus formation, thereby reversing the negative effects of a high-fat diet. These findings suggest the potential benefits of metformin in tumor surveillance.

### 6.4. HCC Cells Transplantation

Xenograft (also known as xenotransplantation) models have emerged as invaluable tools for studying various aspects of liver cancer, thus offering valuable insights into tumor growth, metastasis, and therapeutic responses. Among the diverse available animal models [185,186], zebrafish have gained prominence due to their distinct characteristics and applicability in cancer transplantation studies involving human hepatocellular carcinoma (HCC) cells [187,188].

Mouse xenograft models have long been the gold standard for cancer research. Although mice provide a more anatomically and physiologically appropriate situation for human cancer cells, they also have certain limitations. For example, immunodeficient mice (SCID or athymic nude) have been used to reduce immune rejection [189,190]. Another factor is the time required for tumor development and the costs associated with maintaining a large number of animals to obtain statistically significant results. In addition, the high-resolution live imaging of mice requires specialized techniques, is often invasive, and can interfere with the natural tumor microenvironment and cause artifacts [191].

Zebrafish offers unique advantages over traditional mouse models for the study of liver cancer. Zebrafish, especially in the early developmental stages, possess several characteristics that facilitate liver cancer transplantation. The lack of an adaptive immune system in zebrafish larvae [192] simplifies the transplantation process by minimizing immune rejection, thereby allowing for the successful engraftment of human liver cancer cells. The small size and transparency of zebrafish larvae provide an unprecedented advantage by allowing for noninvasive, high-resolution live imaging. This feature is critical for monitoring tumor growth, metastasis, and the response to anticancer agents at a cellular resolution.

Furthermore, the genetic tractability of zebrafish allows researchers to label specific cell populations with fluorescent proteins, thereby enabling lineage tracing and the analysis of cellular behavior. In addition, the ease of chemical screening and high-throughput approaches in zebrafish accelerate the identification of potential anticancer agents.

HCC is a rapidly growing tumor with a high metastatic potential. Epithelial–mesenchymal transition (EMT) plays a crucial role in the metastatic progression of various cancers, including liver cancer. The loss of cell–cell adhesion and increased cellular motility resulting from EMT enables cancer cells to detach from the primary tumor site. Subsequently, these cells can invade surrounding tissues. The transcription factor Twist, which is associated with EMT, drives metastasis [193]. Elevated Twist expression correlates with poor survival in cancer patients [194]. Zebrafish larvae using the tamoxifen-responsive *Twist1a-ERT2* transgene induce hepatic EMT. In addition, *Twist1a-ERT2/Xmrk* double-transgenic zebrafish [179] show EMT and subsequent spread and dissemination within days, thereby facilitating rapid in vivo drug screening for anti-metastatic compounds.

In vivo drug screening in *Twist1a-ERT2/Xmrk* double-transgenic zebrafish revealed that adrenosterone suppresses hepatic cell dissemination by inhibiting hydroxysteroid dehydrogenase 1 (HSD11β1). This effect was verified in a zebrafish larval xenograft model using a metastatic human HCC cell line. The inhibition of HSD11β1 leads to the re-expression of epithelial marker genes and the reversal of mesenchymal features via downregulation of the transcription factors Snail and Slug. This suggests that *Twist1a-ERT2*-induced EMT alone is insufficient for cell dissemination. *Xmrk*-driven processes are essential for dissemination [179].

In another study, a zebrafish xenograft model using the *achesb55* mutation was established to study liver cancer. The role of acetylcholinesterase (ACHE) as a prognostic marker has been recognized in liver cancer [195]. This mutation led to larger tumors and increased growth, thus implying a potential connection between the host *ache* deficiency and tumor proliferation. This model offers insights into tumor behavior and drug screening [196].

The opacity of adult zebrafish is a limitation in live imaging. However, the recent advent of the Casper zebrafish model [197], which is translucent, provides a solution for conducting high-resolution live imaging studies of HCC tumors even at mature stages [186]. These models will allow researchers to gain a comprehensive understanding of the biology of liver cancer and develop therapeutic approaches to improve the understanding of this complex disease.

## 7. NAFLD and NASH

NAFLD and its more severe condition, NASH, have emerged as common liver disease worldwide in the past few decades, mainly because of the increase in overweight and obesity rates [198]. When NAFLD progresses to NASH, it can progress to advanced liver fibrosis, cirrhosis, and hepatocellular carcinoma (HCC). In 2019, the global prevalence of NAFLD was 29.8% in the total population [199]. Despite the significant medical demand, there is currently no specific drug available for the clinical treatment of NAFLD/NASH owing to its complex pathogenesis.

### 7.1. Limitations of Rodent Models

Rodent models are commonly used for in vivo studies of NAFLD pathogenesis. To induce NAFLD, rats and mice were fed a methionine- and choline-deficient (MCD) diet, which contained high sucrose and fat without methionine and choline. This method is highly effective for inducing steatosis and fibrosis. However, MCD-diet-fed animals showed weight loss and did not show insulin resistance, which has been recognized in patients with NAFLD [200]. Mice fed a high-fat diet (HFD) develop obesity, insulin resistance, and hepatic damage. However, it takes more than 16 weeks for the animals to develop NASH, and they do not show as severe liver damage as those in the MCD model [201]. Therefore, additional animal models that mimic the pathogenesis of NAFLD are required.

### 7.2. Diet-Induced Model

Zebrafish can easily develop hepatic steatosis in both adult and larval stages, and several dietary-induced models have been established. A high-fat diet (HFD), a high-cholesterol diet, and a combination of the two have been used to create NAFLD models at the larval stage [202,203]. Fructose treatment can be used to develop a NASH model in zebrafish larvae [204,205,206]. Notably, gene changes in zebrafish larvae treated with high fructose increased inflammation, oxidative stress, and ER stress more than those in the high-cholesterol or extra-feeding diet models [204]. Further detailed analyses are required to establish the appropriate NAFLD and NASH models.

### 7.3. Chemical-Induced Model

The treatment of zebrafish with particular chemical compounds can provide a valuable model for studying NAFLD. These models provide insights into the molecular mechanisms of NAFLD and potential therapies, and they enable researchers to investigate disease progressions and their impacts on liver health. In this section, we describe several compounds that induce NASH in zebrafish models.

Tributyltin (TBT) is an organostannic compound used in various industrial applications, including as a stabilizer in plastics and marine antifouling paints. Owing to its high level of toxicity and potential to disrupt aquatic ecosystems, TBT has been identified as an environmental contaminant [207]. After 90 days of exposure to TBT, adult zebrafish exhibited yellowish livers with lipid droplets resembling steatosis. Molecular analysis showed that TBT increased the expression of genes related to lipid processes and induced cell damage, thus suggesting that TBT contributes to hepatic steatosis via altered gene activity and cellular stress [208]. Bisphenol S (BPS) is a chemical commonly used in consumer products [209]. Long-term exposure of zebrafish to BPS, ranging from 3 to 120 days after fertilization, leads to increased activity of the liver enzymes and lipid levels, thereby causing apoptosis and fibrosis. These findings provide evidence that exposure to BPS promotes progression from simple steatosis to steatohepatitis in zebrafish. Additionally, BPS activates the PERK-ATF4a pathway, which results in endoplasmic reticulum (ER) stress and accompanying inflammation [210]. Thioacetamide (TAA) is commonly used in laboratory studies to induce liver damage and to mimic conditions similar to those of NAFLD [211]. The zebrafish model treated with TAA can be used to study NAFLD and to evaluate potential therapies [212]. The chemical-induced NAFLD model provides insights into disease mechanisms and potential treatments, although it does not fully replicate human conditions.

### 7.4. Genetically Modified Model

NAFLD has been explained by the two-hit hypothesis, which is divided into the first stage of lipid accumulation in hepatocytes due to the inflow of fatty acids from lipolysis in white adipocytes and de novo synthesis from fructose and glucose in the liver, as well as into the second stage of the progression of inflammation and fibrosis. However, the multiple-hit hypothesis, which suggests that several factors are simultaneously involved in a disease, is widely accepted [213]. These factors include insulin resistance, mitochondrial dysfunction, endoplasmic reticulum (ER) stress, hormones secreted from adipose tissue, gut microbiota, and genetic and epigenetic factors [213]. To clarify the complicated pathophysiology of NAFLD, it would be helpful to use genetically modified models, such as gene-targeted or transgenic expression models. Zebrafish are relatively easy to manipulate genetically, and many transgenic and genetically modified zebrafish models of NAFLD/NASH have been reported [214]. Recently, genome editing using CRISPR/Cas9 technology has increased the accessibility of this approach.

Kulkarni et al. constructed a more efficient NAFLD model by combining a high-fat diet with a liver-specific inflammation-induced transgenic line, thereby demonstrating the possibility of in vivo screening [215]. A more efficient and human-like NASH model can be created by combining dietary models with genetically modified models related to genes, such as insulin resistance, mitochondrial dysfunction, endoplasmic reticulum stress, and inflammation. If such a model could mimic the pathophysiology of NAFLD to NASH in the larval stage, it would lead to a more detailed understanding of the disease and the development of new drugs.

#### 7.4.1. Mutant Model

Recently, zebrafish mutant lines have been developed to phenocopy the key genetic alterations associated with NAFLD. These mutations frequently result in genetic alterations that impair the critical pathways of lipid metabolism, insulin signaling, or inflammation, thereby reflecting aspects of the complex pathophysiology of the disease.

*Trappc11 (foie grans)* is involved in ER-to-Golgi trafficking and exhibits decreased N-linked protein glycosylation (LLO). This is followed by the activation of the unfolded protein response (UPR) and steatosis [117,216].

*Cdipt* plays a key role in phosphatidylinositol (PtdIns) biosynthesis, and it is the primary precursor of phosphoinositides (PIs). The mutation of *cdipt*^hi559^ displays disrupted lipid synthesis, thereby resulting in liver abnormalities resembling NAFLD and chronic ER stress [217].

The *slc16a6a* gene encodes a transporter that moves the ketone body, β-hydroxybutyrate. Disruption of the *slc16a6a* gene leads to the diversion of ketone bodies for fat storage, thus resulting in elevated fat levels in the liver. This discovery reveals the important role of *slc16a6a* in the metabolism of energy during fasting and indicates a possible way to regulate ketone levels in metabolic diseases [218,219].

The *ducttrip (dtp)* mutant is caused by a mutation in the *ahcy* gene, thus functioning as S-adenosylhomocysteine hydrolase. This mutation leads to decreased Ahcy activity and increased S-adenosylhomocysteine (Sah) levels. Elevated Sah levels result in mitochondrial defects and the upregulation of genes involved in lipogenesis, thereby inducing high levels of Tnfα production. Additionally, *AHCY* mutations in patients are linked to hepatic injury and steatosis [220].

#### 7.4.2. Transgenic Model

In addition to mutant models, transgenic zebrafish lines have been developed to modify specific genes linked to NAFLD pathogenesis either by overexpression or inhibition. These models allow researchers to analyze how gene modifications affect lipid accumulation, inflammation, and fibrosis in the zebrafish system.

The expression of a dominant-negative Fgf receptor through *Tg(fabp10a:dnfgfr1-egfp) ^zf421^* in hepatocytes results in smaller liver sizes and impaired expansion in Tg larvae. At 3 mpf, adult Tg fish presented shorter liver lobes but unexpectedly ballooned hepatocytes, which eventually progressed to hepatic steatosis and cholestasis. The Fgf signaling pathway regulates lipid and bile acid metabolism in the livers of adult zebrafish, thus indirectly affecting hepatocyte growth [221].

Experiments using *Tg(–2.8fabp10a:GFP-Yy1)^two16^* showed that overexpression of the ubiquitous transcription factor Yin Yang 1 (*yy1*) leads to the accumulation of triglycerides (TGs) in the zebrafish liver by inhibiting the expression of the *CHOP-10* (*ddit3*) gene and activating two key transcription factors, *C/EBP-α* (*cebpa*) and *PPAR-γ* (*pparg*), that facilitate adipogenesis. This process resulted in the development of liver steatosis, oxidative stress, and lipotoxicity [222].

In *Tg(–2.8fabp10a:Tetoff-Cb1r-2A-EGFP)^two23^*, the cannabinoid receptor 1 (*cb1r*) can be conditionally expressed using a Tet(off) transgenic system. *Cb1r* expression in zebrafish results in increased hepatic lipid accumulation, whereas its suppression reduces lipid accumulation. Adult Tg transgenic fish exhibit elevated lipid content and steatosis. Liver histology confirmed that CB1R induced the stimulation of the lipogenic transcription factor SREBP-1c, and its target enzymes enhanced fatty droplet accumulation, thereby promoting de novo fatty acid synthesis [223].

The excessive expression of activating transcription factor 4 (*atf4*) in zebrafish utilizing the Tet-off transgenic line, *Tg(–2.5actb1:Tetoff-Atf4-2A-mCherry)^zf2124^*, leads to the occurrence of hyperlipidemia and hepatic steatosis. ATF4 boosts lipid biosynthesis-related genes and UPR activity, thereby promoting lipogenesis and adipogenesis. Additionally, the overexpression of *atf4* resulted in an HFD-induced NASH-like phenotype [224]. For more detailed information on the key variants discovered thus far and the transgenic models discussed here, we recommend the following paper [225].

### 7.5. Non-Obese NAFLD

In addition to obesity, NAFLD exhibits a non-obese pattern. It has been estimated that 19.2% of patients with NAFLD are lean, and 40.8% are nonobese [226]. Jung et al. reported differences in the circulating lipidomic profiling in obese and nonobese NAFLD patients [227]. The interleukin 6 (*il-6*) overexpression transgenic model is a diet-independent zebrafish NAFLD model that induces chronic inflammation in the liver by specifically overexpressing IL-6. The livers of IL-6-overexpressed zebrafish larvae accumulated more saturated and monounsaturated triacylglycerols and showed a reduction in polyunsaturated triacylglycerol species that was reminiscent of patients [228]. A model that closely resembles NAFLD, not only in pathology but also in fatty acid composition, could be a powerful tool for screening potential therapeutic agents.

## 8. Cholestasis

Bile is a digestive fluid primarily composed of bile acids and bile pigment (bilirubin), which facilitates fat absorption in the intestine [229]. Bile acids are synthesized from cholesterol in the hepatocyte, accumulated in the gallbladder through the bile ducts, and are transported to the intestine as needed [230]. Cholestasis refers to a condition in which the flow of bile is obstructed, either due to the failure to excrete bile acids synthesized in the liver parenchyma or the obstruction of the biliary tract. If left untreated, the retention of bile acids in the liver and serum can lead to symptoms such as jaundice and generalized pruritus (itching all over the body). Prolonged cholestasis can result in fibrosis of the liver parenchyma, which often progresses to end-stage liver disease with life-threatening complications. The inadequate secretion of bile acids into the intestinal tract leads to decreased fat absorption, malnutrition, and deficiencies in fat-soluble vitamins. External factors, including viral infections, certain medications, and hormonal changes, account for 75% of cholestasis cases, whereas the remaining 25% are attributed to genetic disorders [231]. Currently, mice are the primary animal model used to study the mechanisms of cholestasis and to test the effectiveness of new therapeutic candidates. However, although genetically engineered mice carry disease genes discovered in humans, they may not faithfully replicate the characteristics of human cholestatic liver disease because of species differences [232,233]. Therefore, it is crucial to incorporate multiple animal models to develop a comprehensive understanding of cholestasis pathogenesis.

The zebrafish biliary system consists of intrahepatic and extrahepatic bile ducts, which are closely similar to the human biliary system. Intrahepatic canaliculi transport bile from hepatocytes to larger extrahepatic ducts, thus eventually leading to the gallbladder and intestine [110]. The hepatopancreatic duct connects the liver and pancreas to the intestine, thereby facilitating the transport of bile and digestive enzymes. This highly conserved biliary structure makes zebrafish a suitable model for studying cholestatic conditions, although some architectures have differences (see Section 2.1).

Cholestasis can be induced in zebrafish using diverse strategies, including chemical exposure and genetic manipulation. In fact, various zebrafish models of cholestasis have been reported: Biliary atresia [234,235], Alagille syndrome [6], PFIC2 [123], arthrogryposis–renal dysfunction–cholestasis syndrome [236,237], North American Indian childhood cirrhosis [238], and choledochal cysts [117].

This section will focus on the study of three models of cholestasis, biliary atresia, Alagille syndrome (AGS), and progressive familial intrahepatic cholestasis (PFIC2) using zebrafish, which have been relatively well studied in detail using techniques such as chemical induction and genetic modification.

### 8.1. Biliary Atresia

Biliary atresia (BA) is characterized by inflammation and the destruction of the extrahepatic bile ducts, thereby resulting in obstructive cholestasis [145]. The neonatal mouse model infected with rotavirus (RRV) is commonly used to study BA, as it develops extrahepatic obstruction and an inflammatory response similar to that in human patients within one week of infection [239]. Furthermore, researchers have employed pharmacologically and genetically engineered zebrafish to validate the susceptibility factors and molecular mechanisms discovered through the genome-wide association analysis (GWAS) of patients with BA. These studies have revealed the involvement of DNA hypomethylation as a pathological mechanism leading to cholestasis [235,240]. Notably, zebrafish experiments using *Dysphania glomulifera*, a plant known to induce biliary-atresia-like symptoms in lambs, led to the identification of biliatresone. This compound specifically induces oxidative stress in zebrafish bile duct cells (Figure 3).

Through RNA sequencing in biliatresone-treated zebrafish larvae, researchers identified a mechanistic basis for extrahepatic cholangiocyte (EHC) injury in the redox stress response, specifically in glutathione (GSH) metabolism and the nuclear factor erythroid 2-related factor/Kelch ECH associated protein 1 (Nrf2/Keap1) pathway. Mass spectrometry analysis supported the finding that GSH levels in the EHCs are reduced during the onset of biliary injury. Further experiments using *Tg(ef1a:Grx-roGFP)*, a redox-sensitive GFP biosensor, confirmed the oxidation of GSH in the EHC. Furthermore, GSH supplementation has been found to suppress EHC injury [241]. This highlights the significance of GSH in biliatresone-induced toxicity, which has been observed in other models such as mice and EHC spheroids [242,243]. The reduction of GSH by biliatresone reportedly leads to a decrease in *Sox17* expression via Wnt and Notch signaling, thereby causing toxicity in the EHCs [244]. Additionally, the selective toxicity of biliatresone in the EHCs may be explained by the fact that biliatresone reduced GSH levels in both intrahepatic cholangiocytes (IHCs) and in the EHCs. However, IHCs may be affected by the GSH from hepatocytes that are resistant to biliatresone. Biliatresone has been identified by chemical screening using zebrafish, and the mechanism of action and role of GSH in EHC toxicity has been revealed in detail.

These findings provide valuable insights into the mechanisms underlying the development of biliary atresia and highlights the potential of zebrafish as a model for further research [127].

Currently, there is a lack of evidence regarding the direct exposure of pregnant women to biliatresone. However, there are likely to be other xenobiotics that may have a similar effect. These studies offer a new perspective on the mechanistic basis of BA pathogenesis, which can help to clarify and prevent the cause of the disease.

### 8.2. Alagille Syndrome (AGS)

Alagille Syndrome (AGS) is a genetic disorder that is characterized by chronic cholestasis. It is caused by mutations in *JAG1* or *NOTCH2*, which play crucial roles in the development of various body systems. Most patients with AGS possess dominant mutations in the *JAG1* gene. Studies on mice have demonstrated that mutations in *Jag1* and *Notch 2* genes lead to similar organ defects. Additionally, in zebrafish, the suppression of these genes results in the altered development of the biliary system, thus effectively modeling the characteristics of AGS. These findings confirm the conserved role of Notch signaling in liver development among vertebrates and highlight the significance of zebrafish as a valuable model for studying human biliary system disorders [6,245,246].

### 8.3. Progressive Familial Intrahepatic Cholestasis (PFIC)

The bile acid transport pump BSEP, encoded by the *ABCB11* gene, plays a crucial role in regulating the efflux of bile acid through capillary bile ducts on the surface of hepatocytes, thus acting as a rate-limiting factor for bile secretion [247]. In humans, congenital abnormalities in the BSEP give rise to progressive familial intrahepatic cholestasis type 2 (PFIC2), which is a condition characterized by severe and progressive cholestasis starting in infancy, thus leading to subsequent liver failure and progression to liver cancer [248]. Although extensive research on BSEP function has focused on cultured cells, it is important to note that hepatocytes in planar cultures do not fully replicate the complex environment of the native liver, thus potentially affecting *BSEP* expression and function. Animal models have revealed that *Bsep*-deficient mice are able to maintain bile flow and experience only mild biliary stasis due to the compensatory role of another bile salt transporter called MDR1 (multidrug resistance protein) [232]. However, zebrafish lacking the *bsep* gene exhibit liver damage, as is observed in PFIC2 patients, and have a significantly reduced lifespan, thus serving as the first animal model demonstrating the nearly complete inhibition of bile excretion resulting from the complete loss of Bsep function. In zebrafish lacking *bsep*, *mdr1*, which is normally localized to the plasma membrane, including the capillary bile ducts, was found to migrate to the cytoplasm. A similar phenomenon was observed in human liver tissues lacking the *BSEP*. Conversely, unlike humans and zebrafish, *Bsep*-deficient mice display elevated *Mdr1* expression and localization in the capillary bile ducts, thus suggesting a milder phenotype. Notably, studies conducted in zebrafish have unveiled a novel pathogenic mechanism underlying PFIC2 and have proposed a potential therapeutic strategy for clinical trials. These studies demonstrated that the activation of autophagy through the mammalian target of rapamycin (mTOR) inhibitor rapamycin restored bile excretion function and improved survival in *bsep*-deficient zebrafish, which correlated with the restoration of tubular Mdr1 localization [123]. Pham et al. investigated the causes of cholestasis in children by analyzing the genomes of patients with chronic cholestasis [249]. Their focus was on genes that had not been previously associated with liver disease, and they validated their findings using zebrafish and mouse models. Through the gene sequencing of 93 children with cholestasis and normal levels of γ-glutamyl transferase, they identified a rare pathogenic variant of the *ABCC12* gene. Further analysis revealed additional rare variants of the *ABCC12*, which encodes a protein known as multidrug resistance-associated protein 9 (MRP9). Zebrafish carrying a mutation in the *abcc12* gene exhibit impaired bile duct development and increased apoptosis of the cholangiocytes [249]. Similarly, *Abcc12*-deficient mice showed abnormalities in bile duct formation and increased liver damage when exposed to cholic acid [249].

## 9. Liver Regeneration

The zebrafish liver possesses an impressive ability for self-renewal, thus making it a valuable model for exploring liver regeneration. Research on zebrafish can offer valuable insights into the basic mechanisms of liver regeneration in humans.

The main model for liver regeneration is partial hepatectomy [250]. Research on adult zebrafish demonstrated that excision of the inferior lobe resulted in liver regrowth and restoration of the original structure, which differs from the compensatory growth seen in mammals [60]. Wnt, BMP, and FGF [251] are essential for the signaling pathways necessary for normal liver regeneration. In particular, the Wnt signaling pathway has been analyzed in detail. Liver regeneration was inhibited by reduced Wnt signaling activity in heat-shocked transgenic *Tg(hsp70:dnTCF)* zebrafish. On the other hand, it was found through the analysis of *Tg(hsp70:wnt8)* transgenic fish and *apc* mutants that activating Wnt signaling promotes liver regeneration. This conclusion was confirmed through the analysis of mice with *Apc* mutations [46].

Liver regeneration is a crucial aspect in the treatment of drug-induced liver injury. Acetaminophen (APAP), a drug with sedative and antipyretic properties, is a leading cause of drug-induced acute liver failure worldwide [252]. N-acetylcysteine (NAC) has FDA approval and is used as an antidote (antioxidant) for APAP. However, the use of NAC has some limitations. Therefore, new treatment options are required. Hepatocellular necrosis occurred in the zebrafish model of APAP, which was similar to that observed in humans. Researchers identified prostaglandin 2 (PGE2) using a compound screening approach in the zebrafish model of APAP. Combining PGE2 and NAC in the APAP zebrafish model resulted in liver regeneration, thereby leading to an improvement in the liver condition and prolonged survival. Furthermore, the effect of liver regeneration in PGE2 was found to be mediated by the Wnt signaling pathway [132].

Th inhibition of S-nitrosoglutathione reductase (GSNOR), which is involved in nitric oxide signaling, activates the Nrf2-mediated oxidative stress response pathway. The synergistic effects of the GSNOR inhibitor (N6547) and the antioxidant effect of NAC have been reported to promote liver regeneration in the APAP zebrafish model. GSNOR-deficient mice are also been shown to be resistant to APAP-induced liver injury [253].

Another study using forward genetic screening revealed that zebrafish with a deficiency of *uhrf1* [254] or *top2a* [255,256] have a reduced capacity for liver regeneration.

Recent advancements in zebrafish liver regeneration have included the use of nitroreductase (NTR) to mediate hepatocyte ablation. NTR can transform metronidazole (MTZ) into a cytotoxic metabolite that selectively eliminates particular cells upon exposure to a substrate. The methodology was inspired by gene therapy used for cancer treatment [257]. Two groups pioneered this technique using MTZ to minimize off-target effects [155,258]. Transgenic zebrafish, which express a CFP–NTR fusion driven by the *fabp10a* promoter *(fabp10a:CFP-NTR)*, permit hepatocyte ablation without affecting other cells. Research conducted with this fish line has shown that cholangiocytes (biliary epithelial cells: BECs) can transdifferentiate into hepatocytes during regeneration. The mechanisms involved in this process are *sox9b*, which facilitates BEC transdifferentiation, and *wnt2bb*, which stimulates hepatocyte proliferation [259,260]. In addition, single-cell transcriptomic and high-resolution imaging analyses have revealed that the transformation is regulated by the EGFR–PI3K–mTOR signaling pathway [17]. The MTZ–NTR model was expanded to fibrosis using ethanol-induced stellate cells and laminin–collagen [261]. The model revealed the role of Wnt signaling in regeneration by counteracting Notch signaling. This highlights the importance of MTZ–NTR in liver regeneration.

The unique ability of zebrafish to be imaged in real time allows them to be fully exploited to gain an understanding that will facilitate liver regeneration.

## 10. Conclusions

A comprehensive understanding of liver diseases is essential for the development of effective treatments and optimizing patient outcomes. Drug-induced liver injury (DILI) poses a significant challenge in drug development and requires meticulous preclinical evaluation. Hepatocellular carcinoma (HCC) remains a prominent global health concern, which demands ongoing research on its molecular mechanisms and potential therapeutic interventions. Nonalcoholic steatohepatitis (NASH) and nonalcoholic fatty liver disease (NAFLD) have emerged as primary causes of chronic liver disease, thereby highlighting the importance of early detection and lifestyle interventions. Cholestasis, which is characterized by impaired bile flow, requires further exploration to unravel its pathophysiology and to facilitate the development of targeted therapies. As an emerging model organism, zebrafish offer invaluable insights into liver diseases and offer an ideal platform for genetic and pharmacological investigations. Integrating research across these domains would undoubtedly improve our understanding.

A significant limitation of using zebrafish to model human liver diseases is that their adaptive immune system is not fully developed, especially during the larval stage [192,262]. Innate immunity defends early stage zebrafish until adaptive immunity matures after the first month of life, thereby making it difficult to accurately recreate human immune responses. This limitation needs to be considered thoroughly, as it might affect the modeling of immune-system-associated liver diseases and their related inflammatory responses.

The assessment of liver function relies on the measurement of various serum enzymes, such as alanine aminotransferase (ALT) and aspartate aminotransferase (AST). These enzymes are released into the blood when the liver is damaged or under stress. Various attempts have been made to develop techniques to measure these enzymes in zebrafish. However, the methods and amounts of blood (usually up to 20 µL of whole blood per animal) collected without causing significant injury or stress to their bodies remains a challenge [263,264]. Conventional blood collection techniques used in larger animals do not cause injury or stress. The zebrafish requires a significant recovery period of around 1 to 2 weeks to restore its normal hemoglobin levels after drawing a tiny blood sample of just 2 μL. However, these techniques are often unsuitable for zebrafish owing to their small size. Nevertheless, the development of minimally invasive blood collection techniques for zebrafish is crucial to obtain reliable and appropriate samples.

Zebrafish hepatocytes exhibit physiological differences compared to their human counterparts. Differences in hepatocyte metabolism, drug metabolism, and detoxification pathways can affect the responses of zebrafish to various hepatotoxic agents and influence the interpretation of experimental results. Careful consideration of these physiological differences is essential to ensure the relevance of zebrafish models for studying human liver diseases.

Although the overall anatomical structure of the zebrafish liver resembles that of humans, there are significant differences in terms of lobular organization, vasculature, and biliary architecture. These anatomical variations can affect the localization of disease processes, the distribution of drug effects, and the progression of liver diseases in zebrafish models compared with humans.

## Figures and Tables

**Figure 1 cells-12-02246-f001:**
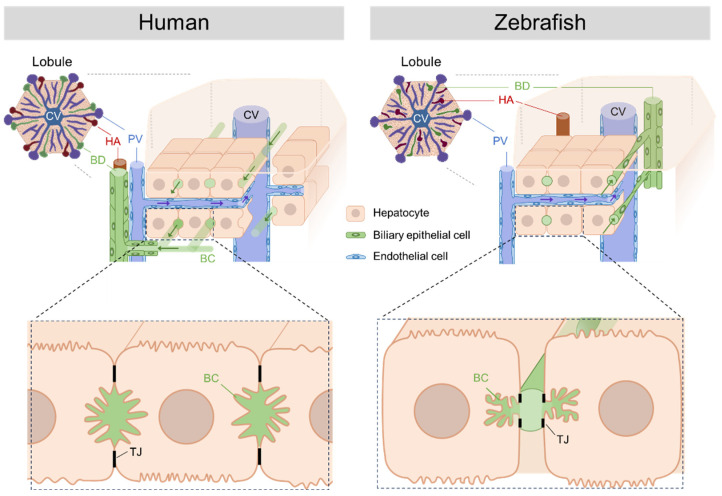
Comparison of the structure of the human and zebrafish liver. The human liver consists of the smallest tissue unit, the polygonal hepatic lobule. The zebrafish liver is thought to have a structure similar to that of the human liver lobule, although it is not structurally less distinct than that of humans. In humans, intrahepatic bile ducts composed of cholangiocytes (biliary epithelial cells) are restricted to the periportal zone, whereas in zebrafish, they are randomly distributed throughout the hepatic lobule. In humans, bile synthesized in hepatocytes is transported through bile canaliculi and microvilli-lined ducts between the hepatocytes connected by tight junctions and then through intrahepatic and extrahepatic bile ducts. In contrast, zebrafish transport bile from hepatocytes through preductules/ductules, which form a mesh-like network with bile canaliculi, and then through the intra- and extrahepatic bile ducts. The liver organ structures include the central vein (CV); portal vein (PV); hepatic artery (HA); bile duct (BD); bile canaliculi (BC); and the tight junction (TJ). The figure presents these structures and was created with BioRender.com (accessed on 26 August 2023).

**Figure 2 cells-12-02246-f002:**
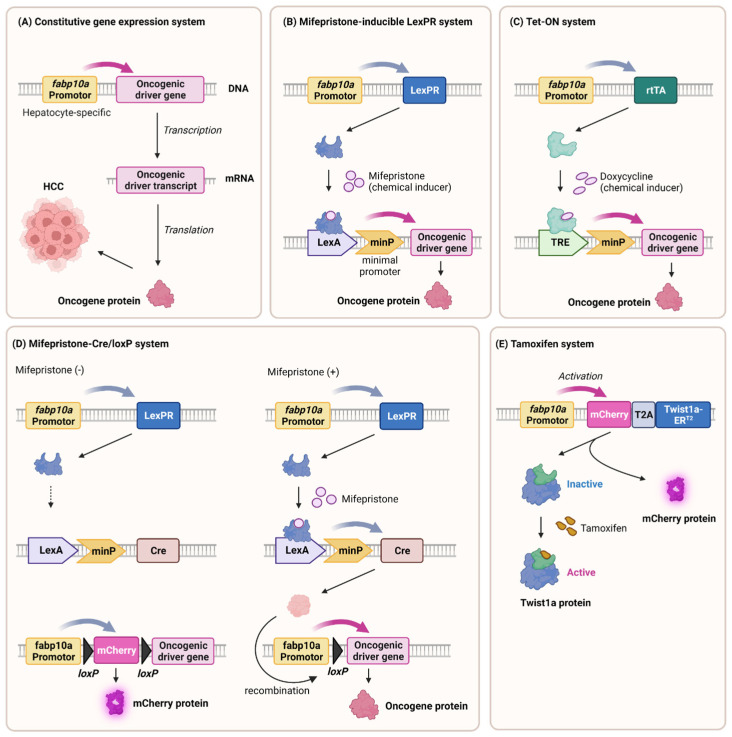
Generation of HCC transgenic zebrafish models. The strategic integration of oncogenic driver genes downstream of the hepatocyte-specific gene expression promoter (*fabp10a*) has been shown to significantly promote the development of hepatocellular carcinoma (HCC), which involves the following: (**A**) a constitutive gene expression system, (**B**) a mifepristone-inducible LexPR system (**C**) a Tet-on system (**D**) a mifepristone–Cre/loxP system (**E**) and a tamoxifen–inducible system. The list provides examples of typical oncogenic driver genes that have been implicated in the induction of HCC in zebrafish. The figure was created with BioRender.com (accessed on 27 August 2023).

**Figure 3 cells-12-02246-f003:**
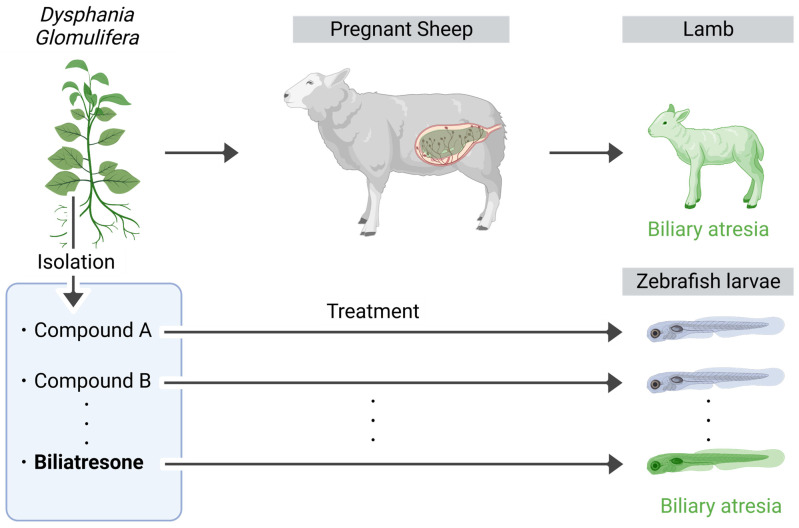
Identification of biliatresone causing biliary-atresia-like symptom. During the dry season, pregnant sheep rely on an Australian native plant as a major component of their diet due to the limited availability of other vegetation. Notably, consumption of this plant leads to the development of biliary atresia in newborn lambs. Significantly, by administering a compound derived from this plant to zebrafish larvae, researchers successfully identified biliatresone as the primary compound responsible for biliary atresia. The figure above was created with BioRender.com (accessed on 26 August 2023).

**Table 2 cells-12-02246-t002:** Summary of DILI zebrafish models.

Classification	Drug	Concentration (µM)	Developmental Stage [Time Duration]	Short Outcome	Ref
Analgesics	Acetaminophen (APAP)	1000–10,000	Adult (3 months) [12–72 h]	Liver size (↓) Necrosis GSH (↓), ALT (↑) Transferrin (↓), LDH (↑)	[134]
		2500–25,000	Larvae (72 hpf) [48 h]		
Antibiotics	Tetracycline (TET)	0.00225–0.225	Juvenile (60 dpf) [30 days]	Lipid vacuoles (↑) Triglyceride (TAG) (↑) Lipid Metabolism (↓)	[136]
		400–4500	Larvae (72 hpf) [48 h]	Liver size (↓) Lipid Metabolism (↓)	[137]
	Erythromycin (ERY)	500–5000	Larvae (72 hpf) [48 h]	Liver size (↓) Lipid Metabolism (↓)	[137]
Nonsteroidal anti-inflammatory drugs	Aspirin [acetylsalicylic acid (ASA)]	15–150	Larvae (72 hpf) [48 h]	Liver size (↓) Lipid Metabolism (↓)	[137]
Antiarrhythmic drugs	Amiodarone (AMD)	1.1–10	Larvae (72 hpf) [48 h]	Liver size (↓)—Necrosis	[139,140]
Immunosuppressive drugs	Cyclosporine A (CsA)	2–8	Larvae (72 hpf) [72 h]	Liver developmental defects ROS (↑)	[132]

## Data Availability

Not applicable.

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
