# Peer review of "Zebrafish as a Useful Model System for Human Liver Disease"

_cells, 2023, doi:10.3390/cells12182246_

Round 1

Reviewer 1 Report

In this review article entitled “Zebrafish as a useful model system for human liver disease”, the authors provide a comprehensive and well-updated review of our understanding of the zebrafish model systems for studying human biomedical research, especially with particular focus on acute liver injury, HCC, NAFLD, NASH and cholestasis. This review article would be a valuable collection for the field of translational research using animal model system. However, the references are mostly not accurate and up to date as well as the figures in the manuscript. I recommend that it to be published with revisions as listed below.

1. On page 1, line 20, the sentence “Zebrafish (Danio rerio), a small fish species that has gained popularity as an ornamental fish, was first discovered in the tributaries of the Ganges River in India in 1822” needs to be revised by considering the following article.

Streisinger, G., Walker, C., Dower, N., Knauber, D. & Singer, F. Production of clones of homozygous diploid zebra fish (Brachydanio rerio). Nature 291, 293–296 (1981).

2. On page 2, line 65 through 74, and figure 1, the paragraph should be revised the following article.

Yilin Yao, Jinxing Lin, Ping Yang, Qiusheng Chen, Xiaohong Chu, Cheng Gao, Jianhua Hu. Fine structure, enzyme histochemistry, and immunohistochemistry of liver in zebrafish. Anat Rec (Hoboken) 2012 Apr;295(4):567-576.

In addition, the authors should mention that the zonation of the zebrafish liver is different from that of the human liver.

3. On page 5, and figure 2, the authors described the transgenic models to investigate liver cancer; however, it needs to add a reference and mentions another cancer by oncogenic driver, beta-catenin. The authors should revise by considering the following article.

Juhoon So, Minwook Kim, Seung-Hoon Lee, Sungjin Ko, Daniel A Lee, Hyewon Park, Mizuki Azuma, Michael J Parsons, David Prober, Donghun Shin. Attenuating the Epidermal Growth Factor Receptor-Extracellular Signal-Regulated Kinase-Sex-Determining Region Y-Box 9 Axis Promotes Liver Progenitor Cell-Mediated Liver Regeneration in Zebrafish. Hepatology, 2021 Apr;73(4):1494-1508.

4. Regarding cholestasis from on page 7, the authors should revise based on the biliary structure of the zebrafish liver.

In this review article entitled “Zebrafish as a useful model system for human liver disease”, the authors provide a comprehensive and well-updated review of our understanding of the zebrafish model systems for studying human biomedical research, especially with particular focus on acute liver injury, HCC, NAFLD, NASH and cholestasis. This review article would be a valuable collection for the field of translational research using animal model system. However, the references are mostly not accurate and up to date as well as the figures in the manuscript. I recommend that it to be published with revisions as listed below.

1. On page 1, line 20, the sentence “Zebrafish (Danio rerio), a small fish species that has gained popularity as an ornamental fish, was first discovered in the tributaries of the Ganges River in India in 1822” needs to be revised by considering the following article.

Streisinger, G., Walker, C., Dower, N., Knauber, D. & Singer, F. Production of clones of homozygous diploid zebra fish (Brachydanio rerio). Nature 291, 293–296 (1981).

2. On page 2, line 65 through 74, and figure 1, the paragraph should be revised the following article.

Yilin Yao, Jinxing Lin, Ping Yang, Qiusheng Chen, Xiaohong Chu, Cheng Gao, Jianhua Hu. Fine structure, enzyme histochemistry, and immunohistochemistry of liver in zebrafish. Anat Rec (Hoboken) 2012 Apr;295(4):567-576.

In addition, the authors should mention that the zonation of the zebrafish liver is different from that of the human liver.

3. On page 5, and figure 2, the authors described the transgenic models to investigate liver cancer; however, it needs to add a reference and mentions another cancer by oncogenic driver, beta-catenin. The authors should revise by considering the following article.

Juhoon So, Minwook Kim, Seung-Hoon Lee, Sungjin Ko, Daniel A Lee, Hyewon Park, Mizuki Azuma, Michael J Parsons, David Prober, Donghun Shin. Attenuating the Epidermal Growth Factor Receptor-Extracellular Signal-Regulated Kinase-Sex-Determining Region Y-Box 9 Axis Promotes Liver Progenitor Cell-Mediated Liver Regeneration in Zebrafish. Hepatology, 2021 Apr;73(4):1494-1508.

4. Regarding cholestasis from on page 7, the authors should revise based on the biliary structure of the zebrafish liver.

Author Response

Dear Editor and Reviewer

We greatly appreciate the careful review of our manuscript entitled “Zebrafish as a useful model system for human liver disease,” submitted for publication in Cells.

The reviewers' valuable and constructive feedback on various aspects of the manuscript is greatly appreciated. We have carefully considered the reviewer’s comments and suggestions and have utilized them to enhance the quality of the paper. In the following section, we have responded to each reviewer's concerns and made the necessary revisions according to their recommendations. Our responses to each comment are indicated in red font below.

Response to Reviewer 1 Comments

Point 1: On page 1, line 20, the sentence “Zebrafish (Danio rerio), a small fish species that has gained popularity as an ornamental fish, was first discovered in the tributaries of the Ganges River in India in 1822” needs to be revised by considering the following article.

Streisinger, G., Walker, C., Dower, N., Knauber, D. & Singer, F. Production of clones of homozygous diploid zebra fish (Brachydanio rerio). Nature 291, 293–296 (1981).

Response 1: We appreciate your feedback regarding the manuscript. We have revised and added the text to reflect your suggestion (on page 1, lines 21-28).

Point 2: On page 2, line 65 through 74, and figure 1, the paragraph should be revised the following article.

Yilin Yao, Jinxing Lin, Ping Yang, Qiusheng Chen, Xiaohong Chu, Cheng Gao, Jianhua Hu. Fine structure, enzyme histochemistry, and immunohistochemistry of liver in zebrafish. Anat Rec (Hoboken) 2012 Apr;295(4):567-576.

Response 2.1: Thank you for this comment. We have revised the text to reflect your suggested article (on page 2, lines 65-96, including the figure legend: on page 2, lines 76-84).

In addition, the authors should mention that the zonation of the zebrafish liver is different from that of the human liver.

Response 2.2: We have reflected on your comment regarding liver zonation (on page 3, lines 97-109).

Point 3: On page 5, and figure 2, the authors described the transgenic models to investigate liver cancer; however, it needs to add a reference and mentions another cancer by oncogenic driver, beta-catenin. The authors should revise by considering the following article.

Juhoon So, Minwook Kim, Seung-Hoon Lee, Sungjin Ko, Daniel A Lee, Hyewon Park, Mizuki Azuma, Michael J Parsons, David Prober, Donghun Shin. Attenuating the Epidermal Growth Factor Receptor-Extracellular Signal-Regulated Kinase-Sex-Determining Region Y-Box 9 Axis Promotes Liver Progenitor Cell-Mediated Liver Regeneration in Zebrafish. Hepatology, 2021 Apr;73(4):1494-1508.

Response 3: We appreciate your suggestion and have integrated the reference into the discussion on transgenic models for liver cancer (on page 13, lines 543-614).

Point 4: Regarding cholestasis from on page 7, the authors should revise based on the biliary structure of the zebrafish liver.

Response 4: Thank you for this comment. In response to your feedback, the manuscript includes an expanded section that outlines the biliary structure of the zebrafish liver (on page 20, lines 852-867).

Reviewer 2 Report

The review article by Shimizu et al., entitled “Zebrafish as a useful model system for human liver disease” highlights various aspect zebrafish as a vertebrate model to study pathological mechanisms of human liver diseases. In this review article, authors have summarized different available tools for analysis of zebrafish liver biology and different type of models of human liver diseases based on zebrafish. The review article is interesting and may be useful for understanding the diverse and complex pathophysiology of human liver disease using zebrafish as a model.

I have following Major concerns about the article.

1) In the section 2 of article, authors have discussed a comparative view of liver morphology and function in zebrafish and humans. This section should be followed by a concise subsection of comparative analysis of liver development in zebrafish and humans, which could be important to discuss about the regenerative therapies in the liver diseases.

2) In the section 2 (title- Useful tools for analysis of zebrafish liver. It should be section 3, not 2), authors have discussed about various transgenic lines that specifically label different liver cell type in zebrafish. However other techniques are also available to study the zebrafish liver biology and function. In this section, authors should present a concise and updated review of other technical methods like antibodies for selective labeling of different cell types including subcellular structures, different histological analysis/staining methods to study liver pathology and different kind of biochemical assays available for studying liver functions. This information be represented in the form of a cartoon or table.

3) In the section 3 (title- Drug-induced Liver Injury model. It should be section 4, not 3), authors have mentioned name of certain drugs for which liver injuries have been demonstrated in zebrafish. In this section, authors should first classify the different categories of drugs like, antibiotics, analgesics, non-steroidal anti-inflammatory drugs, environmental contaminants etc followed by specific drug tested in zebrafish with dosing of drugs, time duration and the developmental stage of zebrafish, drugs were tested with short outcome of treatment. A cartoon type of representation would be easily readable.

4) In the section 5 (NAFLD and NASH), authors have given two groups of NAFLD model, diet induced and genetically modified models, here they shall also include other groups like chemical induced NAFLD (Tributyltin, Bisphenol, Thioacetamide, etc). In the genetically modified models, they can further classify two groups, mutant model, and transgenic models. In these models, they should briefly mention about the mechanism also.

5) Model for the alcoholic liver disease is missing in the review article. Authors should include a separate short section of this highlighting the recent development in this field, because a significant number of people are suffering from alcoholic liver diseases.

6) A short section of the liver regeneration in zebrafish should also be the part of this review article, highlighting new development like in tracking liver regeneration in live zebrafish larvae and screening chemicals to identify regulators of liver regeneration.

7) In the conclusions, authors should also mention about the limitations associated with the zebrafish in modeling human liver diseases, like the adaptive immune system is not fully developed at larval stage, development of methods to assay ALT, AST and other liver enzymes, differences in hepatocyte physiology and hepatic anatomy etc.

Minor Concerns.

1) Page 3, line No. 98 development of hepatocellular carcinoma (HCC) models (see Chapter 5 and Figure.3). There is no chapter 5.

2) Page 7, Line No 277, patients (Jung Y, Aliment Pharmacol Ther. 2020) [74]. The interleukin 6 (IL-6) overexpress. Please correct the reference style.

Author Response

Dear Editor and Reviewer

We greatly appreciate the careful review of our manuscript, “Zebrafish as a useful model system for human liver disease,” submitted for publication in Cells.

The reviewers' valuable and constructive feedback on various aspects of the manuscript is greatly appreciated. We have carefully considered the reviewer’s comments and suggestions and have utilized them to enhance the quality of the paper. In the following section, we have responded to each reviewer's concerns and made the necessary revisions according to their recommendations. Our responses are indicated in red font below each comment.

Response to Reviewer 2 Comments

Point 1: In the section 2 of article, authors have discussed a comparative view of liver morphology and function in zebrafish and humans. This section should be followed by a concise subsection of comparative analysis of liver development in zebrafish and humans, which could be important to discuss about the regenerative therapies in the liver diseases.

Response 1: Thank you for this comment. We revised the text to reflect your suggestion (on page 4, lines 139-215).

Point 2: In the section 2 (title- Useful tools for analysis of zebrafish liver. It should be section 3, not 2), authors have discussed about various transgenic lines that specifically label different liver cell type in zebrafish. However other techniques are also available to study the zebrafish liver biology and function. In this section, authors should present a concise and updated review of other technical methods like antibodies for selective labeling of different cell types including subcellular structures, different histological analysis/staining methods to study liver pathology and different kind of biochemical assays available for studying liver functions. This information be represented in the form of a cartoon or table.

Response 2: Thank you for your suggestion. We added other technical methods (on page 10, Table 1).

Point 3: In the section 3 (title- Drug-induced Liver Injury model. It should be section 4, not 3), authors have mentioned name of certain drugs for which liver injuries have been demonstrated in zebrafish. In this section, authors should first classify the different categories of drugs like, antibiotics, analgesics, non-steroidal anti-inflammatory drugs, environmental contaminants etc followed by specific drug tested in zebrafish with dosing of drugs, time duration and the developmental stage of zebrafish, drugs were tested with short outcome of treatment. A cartoon type of representation would be easily readable.

Response 3: We agree with you. We have added a new table (on page 11, Table2).

Point 4: In the section 5 (NAFLD and NASH), authors have given two groups of NAFLD model, diet induced and genetically modified models, here they shall also include other groups like chemical induced NAFLD (Tributyltin, Bisphenol, Thioacetamide, etc).

Response 4.1: Thank you for this suggestion. We have added chemical induced NAFLD model (on page17, lines722-745)

In the genetically modified models, they can further classify two groups, mutant model, and transgenic models. In these models, they should briefly mention about the mechanism also.

Response 4.2: Thank you for your comment. We have revised the text as you suggest (on page 18, lines 768-820).

Point 5: Model for the alcoholic liver disease is missing in the review article. Authors should include a separate short section of this highlighting the recent development in this field, because a significant number of people are suffering from alcoholic liver diseases.

Response 5: Thank you for your important suggestion. We have added the alcoholic liver disease (ALD) model (on page 12, lines 473-501).

Point 6: A short section of the liver regeneration in zebrafish should also be the part of this review article, highlighting new development like in tracking liver regeneration in live zebrafish larvae and screening chemicals to identify regulators of liver regeneration.

Response 6: We have revised the text accordingly (on page 23, lines 962-1011).

Point 7: In the conclusions, authors should also mention about the limitations associated with the zebrafish in modeling human liver diseases, like the adaptive immune system is not fully developed at larval stage, development of methods to assay ALT, AST and other liver enzymes, differences in hepatocyte physiology and hepatic anatomy etc.

Response 7: Thank you. We have addressed your comment by including the limitations of using zebrafish in modeling human liver diseases in our conclusions (on page 24, lines 1028-1056).

Minor Concerns.

Point 1: Page 3, line No. 98 development of hepatocellular carcinoma (HCC) models (see Chapter 5 and Figure.3). There is no chapter 5.

Response 1: We have corrected this as you suggested (see Chapter 6 and Figure 2).

Point 2: Page 7, Line No 277, patients (Jung Y, Aliment Pharmacol Ther. 2020) [74]. The interleukin 6 (IL-6) overexpress. Please correct the reference style.

Response 2: We have corrected this as you suggested (on page 20, lines 825).

Reviewer 3 Report

The manuscript entitled “Zebrafish as a useful model system for human liver disease” by Nobuyuki Shimizu and coworkers reviews how the zebrafish model could contribute to the understanding of liver diseases in human. The manuscript is clear, well presented and comprehensively written. However, to my opinion several important information related to the zebrafish modeling of liver diseases are missing or require further explanation or clarification. Then, the authors should consider the following points to make their manuscript suitable for publication.

Major.

1. Line 87. “… the advent of CRISPR/Cas9 system …. “ is a reduced vision the methodologies having increased the value of the zebrafish model. The contribution of forward genetics, zebrafish mutagenesis screens, TILLING, transgenesis should be described and acknowledged.

2. Section 2, Liver morphology and function in zebrafish and humans. The authors should discuss the zebrafish complement of Cytochrome P450 and drug metabolism in zebrafish liver.

3. Section 2, Liver morphology and function in zebrafish and humans. The authors should also include in this section a brief description of the steps of liver development in zebrafish: endoderm specification, hepatic specification, hepatic/biliary differentiation, hepatobiliary outgrowth and maturation with the time when the steps occur and markers of the events.

4. Section 2, Useful tools for analysis of zebrafish liver, line 82. “… CRISPR/Cas9 genome editing systems” refers here to knock-in approaches based on the CRISPR/Cas9 technology (ref. 24, 25). This should be clarified and distinguished from the use of CRISPR/Cas9 to induce knock-outs. The authors should also give examples (if any) of the application of the CRIPR/Cas9-based knock-in strategy to liver studies.

5. Section 2, Useful tools for analysis of zebrafish liver. I would recommend to the authors to organize the different transgenic lines labeling liver or liver-associated cells (fapb10a, krt18, notch1, flk1, …) in a summary Table.

6. A section outlining the role of zebrafish as a model for alcoholic liver diseases is lacking.

7. A section outlining the role of zebrafish as a model for liver regeneration is missing.

8. Section 4. Hepatocellular carcinoma. A specific paragraph describing the use of the zebrafish larvae in liver cancer cell transplantation should be added.

9. Section 4. Hepatocellular carcinoma, Figure 2. The fig 2 contains mistakes. “dirver” should be “driver”; In the first line (DNA) labeling “Oncogenic driver gene” is acceptable, but on the second line (mRNA), after the transcription “gene” should be changed into “transcript”.

10. Section 4. Hepatocellular carcinoma, Figure 2. Overall, the figure 2 is useless because it represents a strategy that is not really used. In fact, transgenic fish expressing an oncogene under the control of the fapb10a promoter have a very low (if not null) survival rate after 10 weeks of age. For that reason many of the transgenic fish expressing oncogenes are generated using conditional expression strategies (lexPR/LexA, rtTA/TetO, Cre/loxP). These strategies should be added and discussed in the manuscript.

10. Section 4. Hepatocellular carcinoma, Figure 2. The list of oncogenic driver genes is far to be complete.

11. Section 4. Hepatocellular carcinoma, lines 183-185. The list liver cancer zebrafish models should reorganized since it is a mixture of different strategies having generated them: i.e. apc mutants have been generated through the TILLING approach (loss of function mutants); krasG12V mutants have been generated through transgenesis (gain of function mutants).

Also, the authors should discuss the use of CRISPR/Cas9, in the generation of liver cancer models, i.e. dact2 mutants (Kim et al., 2020, BBRC, 524: 190).

Overall, a more detailed presentation of the liver cancer models should be done.

12. Section 6.1, Biliary atresia, line 315. The authors should describe more precisely how the oxidative stress is measured in zebrafish bile duct cells.

13. Section 2, Useful tools for analysis of zebrafish liver. The identification of genes involved in liver disease through genetic screens should be mentioned in the manuscript; i.e. Sadler et al., A genetic screen in zebrafish identifies the mutants vsp18, nf2 and foie gras as models of liver disease. Development, 2005, 132: 3561.

14. In total, the manuscript lists incompletely a number of zebrafish models of liver disease, but explain too poorly how these models contributed to the understanding of the diseases and how they can contribute to the development of novel therapies. Moreover, the limits of the zebrafish model are not clearly stated. This open ways of improvements for the manuscript.

Minor.

1. Reference [7]: In the reference list, the name of the author is “Roosen-Runge, E.”, but not “E., R.-R.”.

2. The legends for fig. 1, 2 and 3 should be rearranged. Lines 65-74 belong to Figure 1 legend and not to the text of the manuscript; Lines 197-201 belong to Figure 2 legend and not to the text of the manuscript; Lines 321-326 belong to Figure 3 legend and not to the text of the manuscript.

3. Line 98. “See Chapter 5 and Figure 3” should be “Paragraph 4 and Figure 2”

4. Line 277. “(Jung Y, Aliment Pharmacol Ther. 2020)” should be removed.

5. Section 2 appears twice: Page 2, “Liver morphology and function in zebrafish and humans” and Page 3, “Useful tools for analysis of zebrafish liver”. This should be modified.

Author Response

Dear Editor and Reviewer

We greatly appreciate the careful review of our manuscript entitled “Zebrafish as a useful model system for human liver disease,” submitted for publication in Cells.

The reviewers' valuable and constructive feedback on various aspects of the manuscript is greatly appreciated. We have carefully considered the reviewers’ comments and suggestions and have utilized them to enhance the quality of the paper. In the following section, we have responded to each reviewer's concerns and made the necessary revisions according to their recommendations. Our responses are indicated in red font below each comment.

Response to Reviewer 3 Comments

Major.

Point1: Line 87. “… the advent of CRISPR/Cas9 system …. “ is a reduced vision the methodologies having increased the value of the zebrafish model. The contribution of forward genetics, zebrafish mutagenesis screens, TILLING, transgenesis should be described and acknowledged.

Response 1: We appreciate your insight and have taken the opportunity to elaborate on the substantial contributions made by forward genetics, zebrafish mutagenesis screens, TILLING, and transgenesis in the field of zebrafish liver research. The revised manuscript now provides further insight into these methods and their significance (on page 5, lines 219-371).

Point 2: Section 2, Liver morphology and function in zebrafish and humans. The authors should discuss the zebrafish complement of Cytochrome P450 and drug metabolism in zebrafish liver.

Response 2: Thank you for your comment. In line with your comment, the revised manuscript now addresses the diversity of Cytochrome P450 enzymes in zebrafish (on page 3, lines 110-138).

Point 3: Section 2, Liver morphology and function in zebrafish and humans. The authors should also include in this section a brief description of the steps of liver development in zebrafish: endoderm specification, hepatic specification, hepatic/biliary differentiation, hepatobiliary outgrowth and maturation with the time when the steps occur and markers of the events.

Response 3: We appreciate your comment and have incorporated an overview of zebrafish liver development (on page 4, lines 139-215).

Point 4: Section 2, Useful tools for analysis of zebrafish liver, line 82. “… CRISPR/Cas9 genome editing systems” refers here to knock-in approaches based on the CRISPR/Cas9 technology (ref. 24, 25). This should be clarified and distinguished from the use of CRISPR/Cas9 to induce knock-outs. The authors should also give examples (if any) of the application of the CRIPR/Cas9-based knock-in strategy to liver studies.

Response 4: Thank you for your comment. In response to your feedback, we updated the manuscript to distinguish between CRISPR/Cas9-based knock-in and knock-out strategies clearly. The revised content explains how knock-in strategies involve the insertion of specific genetic elements, whereas knock-out strategies involve the disruption of target genes (on page 7, lines 319-345).

Point 5: Section 2, Useful tools for analysis of zebrafish liver. I would recommend to the authors to organize the different transgenic lines labeling liver or liver-associated cells (fapb10a, krt18, notch1, flk1, …) in a summary Table.

Response 5: In response to your suggestion, we revised the manuscript to include a summary table (on page 10, Table 1).

Point 6: A section outlining the role of zebrafish as a model for alcoholic liver diseases is lacking.

Response 6: Thank you for your suggestion. We have integrated a new comprehensive section into the manuscript, focusing on the alcoholic liver disease (ALD) model (on page 12, lines 473-501).

Point 7: A section outlining the role of zebrafish as a model for liver regeneration is missing.

Response 7: We appreciate your comment and have added a section discussing the contribution of zebrafish in liver regeneration research to the manuscript (on page 23, lines 962-1011).

Point 8: Section 4. Hepatocellular carcinoma. A specific paragraph describing the use of the zebrafish larvae in liver cancer cell transplantation should be added.

Response 8: Thank you for your suggestion. We have added a section discussing the contribution of zebrafish larvae to liver cancer cell transplantation (on page 16, lines 637-692).

Point 9: Section 4. Hepatocellular carcinoma, Figure 2. The fig 2 contains mistakes. “dirver” should be “driver”; In the first line (DNA) labeling “Oncogenic driver gene” is acceptable, but on the second line (mRNA), after the transcription “gene” should be changed into “transcript”.

Response 9: We appreciate your suggestion and have corrected the text as indicated (on page 14, Figure 2-A).

Point 10: Section 4. Hepatocellular carcinoma, Figure 2. Overall, the figure 2 is useless because it represents a strategy that is not really used. In fact, transgenic fish expressing an oncogene under the control of the fapb10a promoter have a very low (if not null) survival rate after 10 weeks of age. For that reason many of the transgenic fish expressing oncogenes are generated using conditional expression strategies (lexPR/LexA, rtTA/TetO, Cre/loxP). These strategies should be added and discussed in the manuscript.

Response 10: We appreciate your insight and have updated Figure 2 to reflect contemporary approaches to generating transgenic fish for HCC studies.

Furthermore, the paper now includes a section that addresses the prominent use of conditional expression strategies, such as lexPR/LexA, rtTA/TetO, and Cre/loxP (on page 14, lines 561-614).

Point 11: Section 4. Hepatocellular carcinoma, Figure 2. The list of oncogenic driver genes is far to be complete.

Response 11: We have added more oncogenic driver genes, including the most recent [(on page 13, lines 546-551), (on page 15, lines 573-583), and (on page 15, lines 601)].

Point 12: Section 4. Hepatocellular carcinoma, lines 183-185. The list liver cancer zebrafish models should reorganized since it is a mixture of different strategies having generated them: i.e. apc mutants have been generated through the TILLING approach (loss of function mutants); krasG12V mutants have been generated through transgenesis (gain of function mutants).

Also, the authors should discuss the use of CRISPR/Cas9, in the generation of liver cancer models, i.e. dact2 mutants (Kim et al., 2020, BBRC, 524: 190).

Overall, a more detailed presentation of the liver cancer models should be done.

Response 12: Addressing your suggestion, we have reorganized Section 4, and revised the manuscript to incorporate a discussion of the use of CRISPR/Cas9 in the generation of liver cancer models. Specifically, the inclusion of the suggested paper (Kim et al., 2020, BBRC, 524: 190) has been integrated (on page 13, lines 519-615). 

Point 13: Section 6.1, Biliary atresia, line 315. The authors should describe more precisely how the oxidative stress is measured in zebrafish bile duct cells.

Response 13: Thank you for your comment and suggestion. We have added a description of how to measure oxidative stress in biliatresone-treated zebrafish by adding a reference (on page 21, line 892-907).

Point 14: Section 2, Useful tools for analysis of zebrafish liver. The identification of genes involved in liver disease through genetic screens should be mentioned in the manuscript; i.e. Sadler et al., A genetic screen in zebrafish identifies the mutants vsp18, nf2 and foie gras as models of liver disease. Development, 2005, 132: 3561.

Response 14: Thank you for your comment and suggestion. We have revised the text based on your suggested references (on page 20 , line 860-864, ref 117).

Point 15: In total, the manuscript lists incompletely a number of zebrafish models of liver disease, but explain too poorly how these models contributed to the understanding of the diseases and how they can contribute to the development of novel therapies. Moreover, the limits of the zebrafish model are not clearly stated. This open ways of improvements for the manuscript.

 Response 15: Thank you for your valuable feedback regarding our manuscript. In alignment with your feedback, the manuscript now takes a more balanced approach by not only listing zebrafish models of liver disease, but also providing a deeper exploration of their contributions to disease comprehension and therapeutic avenues. Furthermore, a distinct section has been added to articulate the limitations of the zebrafish model, fostering a more comprehensive and informative narrative.

Minor.

Point 1: Reference [7]: In the reference list, the name of the author is “Roosen-Runge, E.”, but not “E., R.-R.”.

Response 1: We have improved this as you suggested (on page 25, lines 1085).

Point 2: The legends for fig. 1, 2 and 3 should be rearranged. Lines 65-74 belong to Figure 1 legend and not to the text of the manuscript; Lines 197-201 belong to Figure 2 legend and not to the text of the manuscript; Lines 321-326 belong to Figure 3 legend and not to the text of the manuscript.

Response 2: We have rearranged the figure legend and text based on your suggestions.

Point 3: Line 98. “See Chapter 5 and Figure 3” should be “Paragraph 4 and Figure 2”

Response 3: We have replaced this sentence as you suggested (on page 9, lines 397).

Point 4: Line 277. “(Jung Y, Aliment Pharmacol Ther. 2020)” should be removed.

Response 4: We have removed this as you suggested (on page 20, lines 825).

Point 5: Section 2 appears twice: Page 2, “Liver morphology and function in zebrafish and humans” and Page 3, “Useful tools for analysis of zebrafish liver”. This should be modified.

Response 5: We modified the manuscript according to your suggestion.

Round 2

Reviewer 3 Report

The authors positively responded to my concerns. As it stands, the quality of revised manuscript is significantly improved.

No more comments.